# Discovery and remodeling of *Vibrio natriegens* as a microbial platform for efficient formic acid biorefinery

Jinzhong Tian [1,2,5] ✉, Wangshuying Deng[1,3,5], Ziwen Zhang[1,3], Jiaqi Xu[4], Guiling Yang[2], Guoping Zhao[1], Sheng Yang [1], Weihong Jiang [1] ✉ & Yang Gu [1] ✉

Formic acid (FA) has emerged as a promising one-carbon feedstock for biorefinery. However, developing efficient microbial hosts for economically competitive FA utilization remains a grand challenge. Here, we discover that the bacterium *Vibrio natriegens* has exceptional FA tolerance and metabolic capacity natively. This bacterium is remodeled by rewiring the serine cycle and the TCA cycle, resulting in a non-native closed loop (S-TCA) which as a powerful metabolic sink, in combination with laboratory evolution, enables rapid emergence of synthetic strains with significantly improved FA-utilizing ability. Further introduction of a foreign indigoidine-forming pathway into the synthetic *V. natriegens* strain leads to the production of 29.0 g·L⁻¹ indigoidine and consumption of 165.3 g·L⁻¹ formate within 72 h, achieving a formate consumption rate of 2.3 g·L⁻¹·h⁻¹. This work provides an important microbial chassis as well as design rules to develop industrially viable microorganisms for FA biorefinery.

Microbial assimilation and conversion of one-carbon (C1) compounds to produce value-added products have attracted significant attention because it represents an important biological manufacturing route using abundant and highly available carbon sources[1]. Formic acid (FA) is a promising C1 feedstock since it can be efficiently derived from $CO_2$ or syngas (mainly CO and $CO_2$) with high selectivity via chemical conversions and further utilized by microorganisms[2]. Additionally, the storage of FA as a liquid is more convenient compared to gases[3]. Thus, FA is a $CO_2$-equivalent carbon source that is more easily handled than $CO_2$.

Native FA-utilizing microorganisms mainly employ the Wood-Ljungdahl pathway (also known as the reduced acetyl-CoA pathway), serine cycle, and reductive glycine pathway for FA assimilation[4]. These pathways all depend on a core FA-fixing module called the tetrahydrofolate (THF) cycle which is initiated by formate-tetrahydrofolate ligase (FTL)[4]. Pyruvate formate-lyase (PFL)-catalyzed pyruvate

formation from formate and acetyl-CoA is another important FA-assimilating approach in organisms[4]. However, native FA-utilizing microorganisms generally grow slowly and cannot efficiently utilize this C1 carbon source[5–17]. Hence, efforts have been made to develop synthetic microorganisms such as *Escherichia coli* through rational genetic modifications and adaptive laboratory evolution, wherein FA assimilation was achieved by reconstructing the THF cycle and reverse glycine cleavage[18–20]. Despite the improved performance of synthetic microorganisms in FA utilization, the efficiency is still much lower than those of traditional carbon sources such as sugars, which is largely due to the native weak FA tolerance and metabolic capacity of these microbial hosts. Thus, the discovery and employment of more suitable microorganisms as a chassis to establish efficient FA bioconversion platforms are needed.

*Vibrio natriegens* is a rapidly growing bacterium with the shortest doubling time (<10 min) of all known bacteria[21,22]. It is also regarded as

[1]CAS-Key Laboratory of Synthetic Biology, CAS Center for Excellence in Molecular Plant Sciences, Shanghai Institute of Plant Physiology and Ecology, Chinese Academy of Sciences, Shanghai 200032, China. [2]Xianghu Laboratory, Hangzhou 311231, China. [3]University of Chinese Academy of Sciences, Beijing, China. [4]ZJU-Hangzhou Global Scientific and Technological Innovation Center, Hangzhou 311215, China. [5]These authors contributed equally: Jinzhong Tian, Wangshuying Deng. ✉e-mail: tianjinzhong@cemps.ac.cn; wjiang@cemps.ac.cn; ygu@cemps.ac.cn

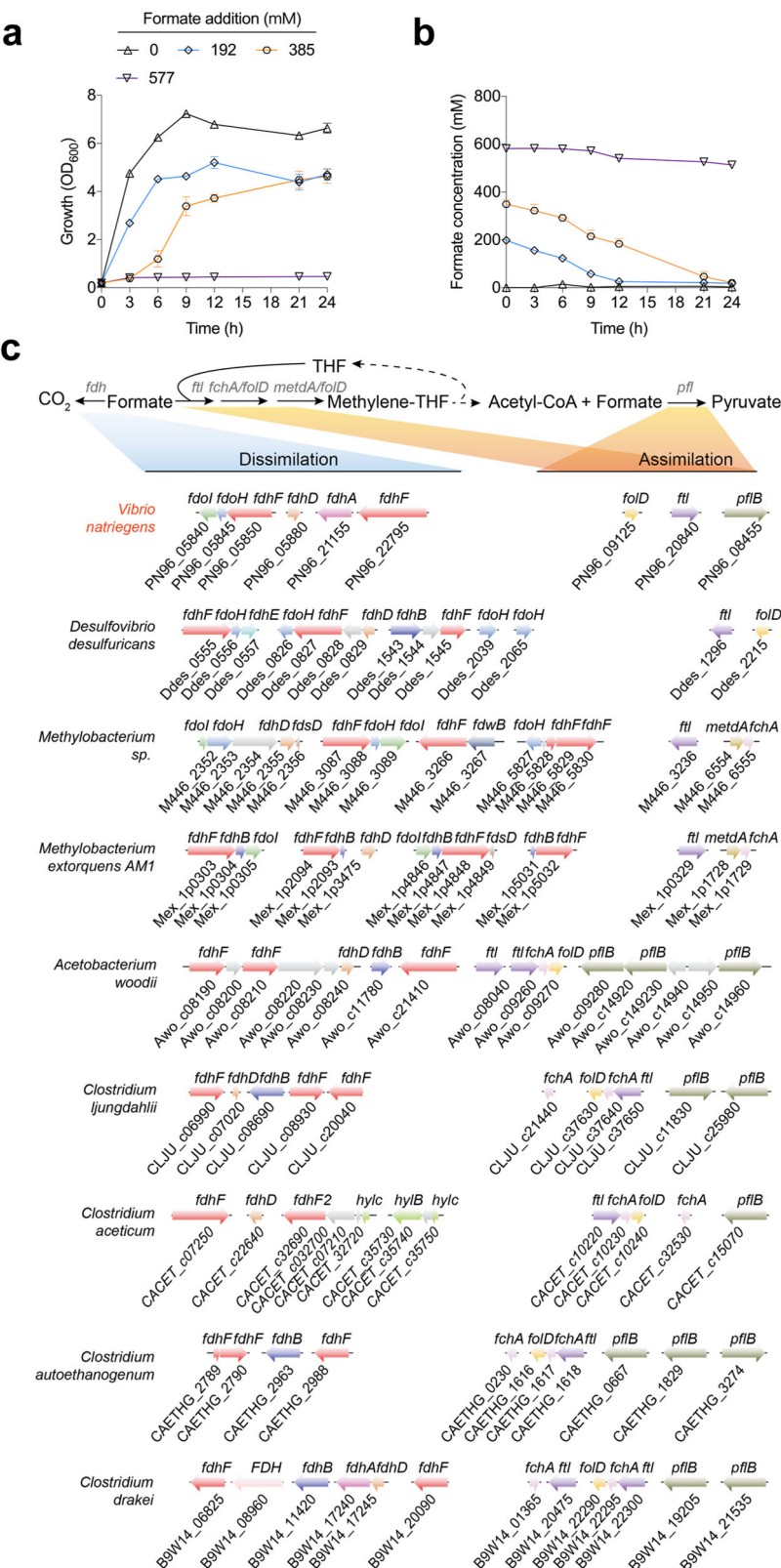

a desirable microbial host for bio-manufacturing[23–25]. *Vibrio natriegens* can grow on diverse substrates including galactose, arabinose, and glycerol[26,27], due to its versatile metabolism. Furthermore, genetically modified *V. natriegens* strains are capable of producing multiple chemicals including amino acids, polymers, and polyols[27–29]. However, its inherent physiological and metabolic characteristics and potential in utilizing C1 resources remain unclear.

In this work, we discover the previously unreported superior tolerance and metabolic capacity of *Vibrio natriegens* to FA and develop this bacterium into a platform microorganism for FA utilization and conversion. Using a combination of bioinformatic, omics, and experimental validation approaches, multiple potentially crucial genes and metabolic pathways associated with FA metabolism in *Vibrio natriegens* are revealed. Based on these findings, we design and remodel this

**Fig. 1 | Comparative analysis of formate metabolic pathways in _V. natriegens._**
**a**, **b** Cell growth (**a**) and formate consumption (**b**) profiles with the supplementation of different amounts of sodium formate (HCOONa·2H$_2$O). Data are presented as the mean ± SD ($n$ = 3 biologically independent samples). Error bars show SDs. **c** Putative genes and gene clusters involved in formate assimilation and dissimilation pathways in _V. natriegens_ and some other representative formate-utilizing microorganisms. Homologous genes are marked with the same colors. _fdoI_ formate dehydrogenase subunit gamma, _fdhB_ formate dehydrogenase (NADP$^+$) beta subunit, _fdoH_ formate dehydrogenase iron-sulfur subunit, _fdhF_ formate

dehydrogenase major subunit, _fdsD_ formate dehydrogenase subunit delta, _fdhA_ formate dehydrogenase (coenzyme F420) alpha subunit, _fdwB_ formate dehydrogenase beta subunit, _FdhD_ formate dehydrogenase accessory protein, _ftl_ formate--tetrahydrofolate ligase, _fchA_ methenyltetrahydrofolate cyclohydrolase, _mtdA_ Methylenetetrahydrofolate dehydrogenase (NADP$^+$), _folD_ bifunctional methylenetetrahydrofolate dehydrogenase/methenyltetrahydrofolate cyclohydrolase, _pflB_ pyruvate formate-lyase, _hylB_ formate dehydrogenase (NAD$^+$, ferredoxin) subunit B, _hylC_ formate dehydrogenase (NAD$^+$, ferredoxin) subunit C. Source data are provided as a Source Data file.

bacterium, combined with laboratory evolution, achieving efficient FA utilization and indigoidine production.

## Results

### Superior tolerance and metabolic capacity of native _Vibrio natriegens_ to formate

_V. natriegens_ has rapid growth, but its tolerance and metabolic capacity to formate are unknown; thus, we first examined its growth and formate consumption in the presence of different formate concentrations. We observed that _V. natriegens_ could grow in the LBv2 medium supplemented with 20 and 40 g·L$^{-1}$ sodium formate (HCOONa·2H$_2$O) (equivalent to 192 and 385 mM formate, respectively) (Fig. 1a), and exhausted all the formate within 24 h (Fig. 1b), although the growth rates and highest biomass gradually decreased with increasing formate concentration (Fig. 1a). At a higher sodium formate concentration (60 g · L$^{-1}$, equivalent to 577 mM), the strain could hardly grow (Fig. 1a), but it still consumed 69 mM formate within 24 h (Fig. 1b). However, when _V. natriegens_ was cultivated in the M9 minimal medium (containing 4 g · L$^{-1}$ glucose), the addition of 288 mM formate (30 g · L$^{-1}$ HCOONa·2H$_2$O) could completely inhibit cell growth (Supplementary Fig. 1a), showing poorer formate tolerance relative to that of the LBv2 medium. At a lower formate concentration (192 mM, equivalent to 20 g · L$^{-1}$ HCOONa·2H$_2$O) in this minimal medium, _V. natriegens_ could grow and consume 95 mM formate within 24 h (3.96 mM·h$^{-1}$) (Supplementary Fig. 1b, c). Taken together, compared with the reported data from other microorganisms, _V. natriegens_ exhibited a superior ability in the consumption of formate (Supplementary Table 1), showcasing great potential in biological FA utilization.

### Genes and metabolic pathways for formate metabolism in _V. natriegens_

Natural FA assimilation reactions in organisms are quite scarce, in which two of the best known are FTL-catalyzed formylation of

tetrahydrofolate (THF) and (PFL)-catalyzed pyruvate formation (Supplementary Fig. 2). Additionally, FA can be oxidized to form CO$_2$ by formate dehydrogenase (FDH), a major FA dissimilation reaction in microorganisms. The _V. natriegens_ genome contains six _fdh_ genes (two _fdhF_s, one _fdhA_, one _fdoI_, one _fdoH_, and one _fdhD_), two THF cycle genes (_ftl_ and _folD_) and one _pfl_ gene, enabling the operation of both FA dissimilation and assimilation reactions mentioned above (Fig. 1c); however, the other two THF cycle genes, _fchA_ and _metdA_, were missing (Table 1). Furthermore, all the candidate genes responsible for the reactions starting from methylene-THF in the reductive glycine pathway and serine cycle were found in _V. natriegens_ (Supplementary Fig. 3a), suggesting that the assimilated formate can be further converted into diverse metabolites. Additionally, energy conservation systems play important roles in microbial growth on organic C1 resources such as formate and methanol[20]. Of note, diverse energy conservation systems were found in _V. natriegens_, most of which had multiple coding genes and gene clusters in the genome (Supplementary Fig. 3b). This may be crucial for the required energy supply in the growth of _V. natriegens_ on formate.

To explore the potential genes and metabolic pathways associated with formate metabolism in _V. natriegens_, we measure the transcriptomic changes with the induction of formate. There were 755 and 606 genes significantly up-regulated and down-regulated, respectively, following formate addition to the culture (Supplementary Fig. 4a). Functional category analysis yielded 28 subsets containing many genes exhibiting significantly altered transcriptional levels (Supplementary Fig. 4b). An obvious positive correlation was detected between qRT-PCR and RNA-seq results for 15 selected genes (Supplementary Fig. 5), suggesting a good quality of the comparative transcriptomic data. We observed that, with formate stress, multiple crucial genes located in the THF cycle, the reductive glycine pathway, and the serine cycle were significantly up-regulated (Fig. 2a). Four (PN96_22795, PN96_05840, PN96_05845, and PN96_05850) out of six FDH-encoding genes were up-regulated with the highest fold change (9.7-fold) occurring for PN96_05850 (Fig. 2a). Furthermore, an 8.9-fold transcriptional increase in the _pfl_ gene which encodes pyruvate formate-lyase occurred (Fig. 2a). These findings indicate that the formate assimilation and dissimilation reactions mediated by these genes may contribute to the formate metabolism in _V. natriegens_. Noticeably, the transcriptional levels of nearly all the genes located in the TCA cycle were significantly up-regulated (Fig. 2a). The TCA cycle is the metabolic hub in cells connecting major nutrients, including sugars, lipids, and amino acids[30,31]. The overall up-regulation of TCA genes indicates that _V. natriegens_ enhanced its fundamental metabolism to meet the energy demand in formate assimilation. In addition, multiple genes encoding ionic transporters or associated with unsaturated fatty acid synthesis and DNA repair were significantly upregulated (Supplementary Fig. 6). It is known that the acid tolerance of microorganisms can be affected by the efficiency of ionic transporters and the ratio of unsaturated fatty acids in cell membrane[32-34]; hence, higher expression of these genes may be crucial for the formate tolerance of _V. natriegens_.

Based on the above analyses, _V. natriegens_ is predicted to metabolize formate through FTL and PFL-catalyzed assimilation reactions as well as an FDH-catalyzed dissimilation reaction (Fig. 2b, upper). To

**Table 1 | Occurrence of genes involved in formate metabolism in _V. natriegens_ and some other representative formate-utilizing microorganisms**

| Microorganisms | Dissimilation | Assimilation | | | | |
|---|---|---|---|---|---|---|
| | _fdh_ | _ftl_ | _fchA_ | _metdA_ | _folD_ | _pfl_ |
| _Vibrio natriegens_ | + | + | | | + | + |
| _Desulfovibrio desulfuricans_ | + | + | | | + | |
| _Methylobacterium sp._ | + | + | + | + | | |
| _Methylobacterium extorquens_ AM1 | + | + | + | + | | |
| _Acetobacterium woodii_ | + | + | + | | + | + |
| _Clostridium ljungdahlii_ | + | + | + | | + | + |
| _Clostridium aceticum_ | + | + | + | | + | + |
| _Clostridium autoethanogenum_ | + | + | + | | + | + |
| _Clostridium drakei_ | + | + | + | | + | + |

The presence of genes for the corresponding functions is indicated with a "+".

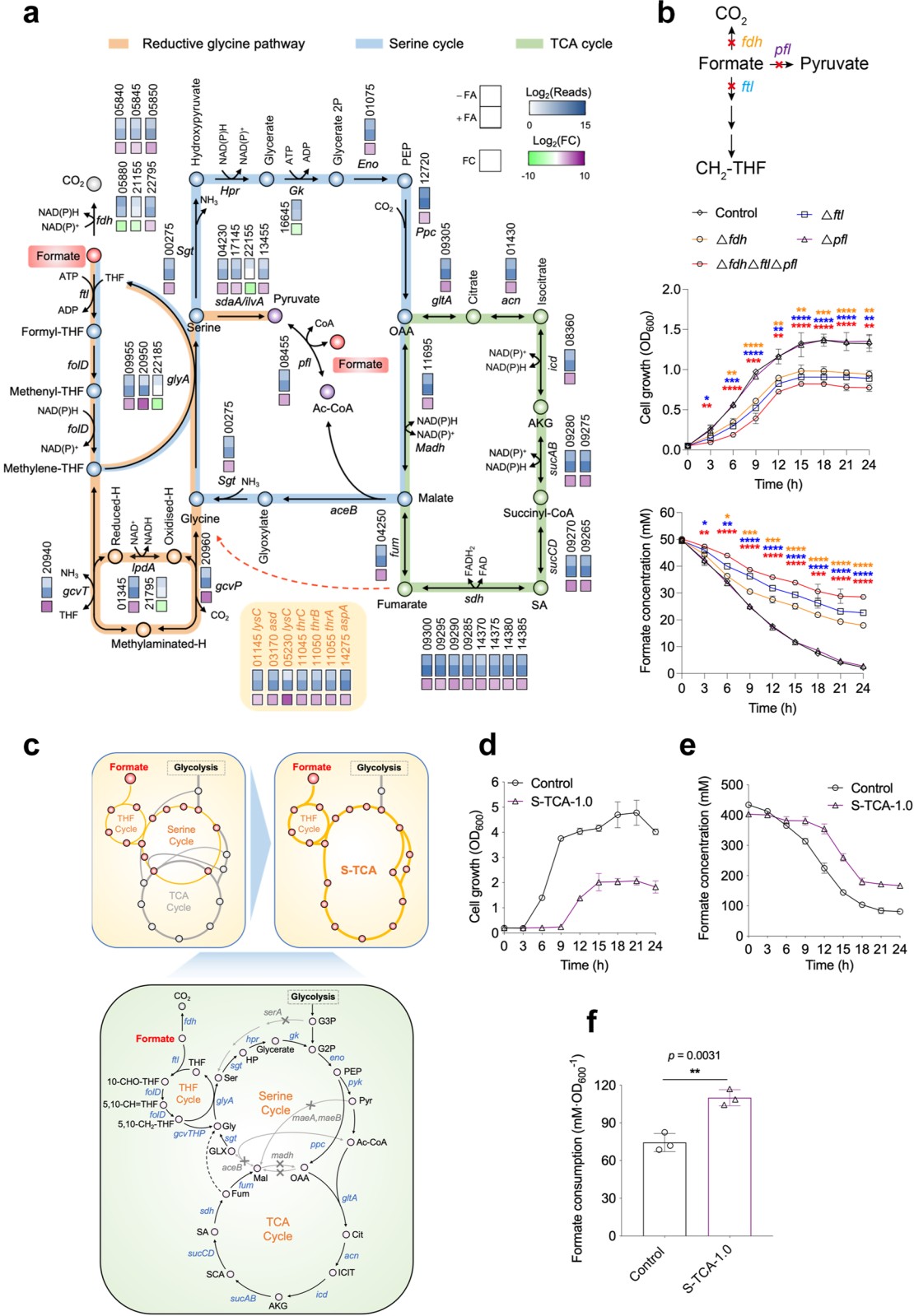

test them, the candidate genes (i.e., *fdh*, *ftl* and *pfl*) responsible for these reactions were separately or simultaneously deleted in *V. natriegens* for phenotypic analysis (Fig. 2b, upper). We observed that the deletion of the *fdh* genes or the *ftl* gene significantly impaired cell growth and formate consumption, indicating that both the formate dissimilation (oxidation) and assimilation pathways mediated by *fdh* and *ftl*, respectively, play roles in the formate metabolism of *V.*

*natriegens* (Fig. 2b, middle and bottom). However, the deletion of the *pfl* gene had no influence on cell growth and formate consumption (Fig. 2b, middle and bottom), although this gene was up-regulated in the presence of formate (Fig. 2a). This finding indicates the low catalytic activity of PFL towards formate in *V. natriegens*. Furthermore, the simultaneous deletion of all the *fdh*, *ftl*, and *pfl* genes led to less formate consumption compared to the separate deletion of the *fdh* or *ftl*

**Fig. 2 | Validation of formate assimilation and dissimilation pathways and reconstitution of formate metabolism in *V. natriegens*. a** The significantly up-regulated or down-regulated genes involved in the tetrahydrofolate (THF) cycle, TCA cycle, serine cycle, pyruvate formate-lyase (PFL) pathway and formate dehydrogenase (FDH)-mediated formate dissimilation pathways with the supplementation of sodium formate in *V. natriegens*. The bottom box indicates fold change as gene expression with FA over gene expression without FA. **b** Influence of blocking the FTL-mediated THF cycle, pyruvate formate-lyase (PFL) pathway and formate dehydrogenase (FDH)-mediated formate dissimilation pathway (upper) on cell growth (middle) and formate metabolism (bottom) of *V. natriegens*. The deleted *ftl* and *pfl* gene were PN96_20840 and PN96_08455, respectively. The *fdh*-deleted mutant was constructed by simultaneous deletion of the six annotated *fdh* genes (PN96_05840, PN96_05845, PN96_05850, PN96_05880, PN96_21155, and PN96_22795). Sodium formate (HCOONa·2H$_2$O) and glucose supplemented in the M9 medium was $5.0 \, g \cdot L^{-1}$ and $4.0 \, g \cdot L^{-1}$, respectively. Data are presented as the mean ± SD ($n = 3$ biologically independent samples). **c** The combined manipulations fused the serine cycle and TCA cycle to create a metabolic sink to promote formate metabolic flux. **d, e** The growth (**d**) and formate consumption (**e**) of the wild type strain and the reconstructed S-TCA-1.0 strain in the LBv2 medium supplemented with $40 \, g \cdot L^{-1}$ sodium formate. **f** Formate consumption per unit biomass of the wild type and S-TCA-1.0 strains in the LBv2 medium supplemented with $40 \, g \cdot L^{-1}$ sodium formate. Data are presented as the mean ± SD ($n = 3$ biologically

independent samples). *fdh* formate dehydrogenase, *ftl* formate--tetrahydrofolate ligase, *folD* bifunctional methylenetetrahydrofolate dehydrogenase/methenyltetrahydrofolate cyclohydrolase, *glyA* glycine hydroxymethyltransferase, *sgt* serineglyoxylate transaminase, *sdaA* serine dehydratase, *ilvA* threonine dehydratase, *Hpr* glycerate dehydrogenase, *Gk* glycerate 2-kinase, *eno* enolase, *ppc* phosphoenolpyruvate carboxylase, *gltA* citrate synthase, *acn* aconitate hydratase 2, *icd* isocitrate dehydrogenase, *sucA* 2-oxoglutarate dehydrogenase E1 component, *sucB* 2-oxoglutarate dehydrogenase E2 component, *sucC* succinyl-CoA synthetase beta subunit, *sucD* succinyl-CoA synthetase alpha subunit, *pfl* pyruvate formate-lyase, *sdh* succinate dehydrogenase, *fum* fumarate hydratase, *madh* malate dehydrogenase, *aceB* malate synthase, *maeA* malate dehydrogenase (oxaloacetate-decarboxylating), *maeB* malate dehydrogenase (oxaloacetate-decarboxylating) (NADP+), *gcvT* glycine cleavage system T protein, *gcvH* glycine cleavage system H protein, *gcvP* glycine cleavage system P protein, *lpdA* dihydrolipoyl dehydrogenase, *pyk* pyruvate kinase, THF tetrahydrofolate, HP hydroxypyruvate, PEP phosphoenolpyruvate, OAA oxaloacetate, AKG alpha-ketoglutaric acid, Cit citrate, ICIT isocitrate, SA succinic acid, SCA Succinyl-CoA, Pyr pyruvate, G2P 2-Phospho-D-glycerat, G3P 3-Phospho-D-glycerate, Mal malate, Fum fumarate, GLX Glyoxylate, Gly glycine, Ser serine. Error bars show SDs. Statistical analysis was performed by a two-tailed Student's *t*-test. *$P < 0.05$; **$P < 0.01$; ***$P < 0.001$; ****$P < 0.0001$ versus the control strain. Source data are provided as a Source Data file.

genes (Fig. 2b, bottom); but interestingly, the mutated strain (with the deletion of all the *fdh*, *ftl*, and *pfl* genes) still maintained partial consumption ability of formate (Fig. 2b, bottom), indicating that these pathways were not completely blocked or there are other unknown approaches contributing to the formate metabolism in *V. natriegens*.

### Designing and creating a metabolic sink to promote formate metabolic flux

The above findings prompted us to reprogram *V. natriegens* to further enhance its formate utilization. To this end, the following basic rationale was adopted: construction of a powerful metabolic sink that is linked to formate assimilation by integrating the TCA cycle and the serine cycle, aiming at increasing the flux of the serine cycle and its upstream THF cycle by the strong pull from the TCA cycle (Fig. 2c, upper). This was achieved by manipulating the following pathways and genes (Fig. 2c, bottom): (i) disruption of three *madh* genes (PN96_06470, PN96_19465, and PN96_11695), two *maeAB* genes (PN96_07295 and PN96_14755), and the *aceB* gene (PN96_10585) to fuse the serine cycle and the TCA cycle, thus forcing the metabolic flux of the serine cycle to go through the TCA cycle; (ii) disruption of the *serA* gene (PN96_00930) to enable the generation of more serine from formate assimilation rather than glycolysis.

The resulting strain S-TCA-1.0 was cultivated with the supplementation of $40 \, g \cdot L^{-1}$ sodium formate (HCOONa·2H$_2$O) (equivalent to 385 mM formate). Compared with the wild type strain, S-TCA-1.0 exhibited impaired growth and less total formate consumption (Fig. 2d, e); but interestingly, its formate consumption per unit biomass was much higher than that of the wild type strain (Fig. 2f), indicating the utility of our metabolic engineering strategy in enhancing the formate metabolic flux of individual *V. natriegens* cells. To confirm that the S-TCA cycle truly contributed to the enhanced formate consumption, we examined the transcriptional differences of all the S-TCA genes between the S-TCA-1.0 and the wild-type strains. As expected, all these genes could be properly expressed in S-TCA-1.0, and moreover, most of them exhibited higher transcription levels in S-TCA-1.0 relative to the wild-type strain (Supplementary Fig. 7). We also assayed multiple metabolites (glycine, serine, hydroxypyruvate, glycerate, phosphoenolpyruvate, α-Ketoglutarate, succinic acid, and fumarate) throughout the S-TCA cycle in the S-TCA-1.0 strain. We found that all these eight compounds could be detected, in which five showed higher concentration in S-TCA-1.0 relative to the wild-type strain, while the other three had no significant differences between the two strains (Supplementary Fig. 8). Our analysis suggests that the S-TCA cycle

exerts its function in the cells of S-TCA-1.0, leading to enhanced formate metabolic flux. If the growth deficiency of S-TCA-1.0 can be remedied, this strain will exhibit advantages over the wild type strain in formate utilization.

### Adaptive laboratory evolution of S-TCA-1.0 leads to a high-performance strain grown on formate

To efficiently overcome the growth limitation of S-TCA-1.0, this strain was subjected to adaptive laboratory evolution (ALE) with gradually increasing sodium formate concentrations to improve its growth under formate stress. The strain was initially cultivated in the LBv2 medium supplemented with approximately $20 \, g \cdot L^{-1}$ sodium formate (HCOONa·2H$_2$O) (equivalent to 192 mM formate) and subcultured once every 12 h (Fig. 3a). Several rounds of subculturing resulted in the biomass of grown cells reaching OD$_{600}$ ~ 1.0. At this point, the formate concentration was increased and a new evolutionary period began. This approach gradually improved the tolerance of S-TCA-1.0 to formate as well as its growth. Finally, an evolved strain (named S-TCA-2.0) capable of growing in the presence $140 \, g \cdot L^{-1}$ sodium formate (equivalent to 1,346 mM formate) was obtained after serial subculturing (Fig. 3b). More importantly, S-TCA-2.0 exhibited exceedingly high formate utilization ability, capable of growing to OD$_{600}$ ~ 1.6 with the consumption of $78.9 \, g \cdot L^{-1}$ sodium formate (equivalent to 759 mM formate) within 24 h following initial supplementation of $85 \, g \cdot L^{-1}$ sodium formate (equivalent to 817 mM formate) in the medium (Fig. 3c), reaching a formate consumption rate of $1.42 \, g \cdot L^{-1} \cdot h^{-1}$ (31.6 mM·h$^{-1}$) which was several times higher than the highest level reported in microorganisms (Supplementary Table 1).

An independent ALE experiment was performed in parallel to improve the formate utilization of the wild-type strain of *V. natriegens*; however, the cellular adaptation of this strain under formate stress was much slower than that of S-TCA-1.0, and enhancement was difficult when the concentration of sodium formate reached $70 \, g \cdot L^{-1}$ (equivalent to 673 mM formate) (Supplementary Fig. 9). These results suggest that the reconstituted formate metabolic pathways in *V. natriegens* (S-TCA-1.0) facilitated the strain evolution to adapt to high formate stress, increasing its formate consumption accordingly.

To better elucidate the genetic basis for the performance of S-TCA-2.0 in using formate, we chose three isolates to sequence their genome. A total of nine mutated genes were found in all the three isolates of S-TCA-2.0 relative to the S-TCA-1.0 strain (Fig. 3d and Table 2). Among them, *rpoS*, *ZapC*, and *actP*, encoding a general stress responsible regulator[35], a division protein[36], and an acetate permease[37],

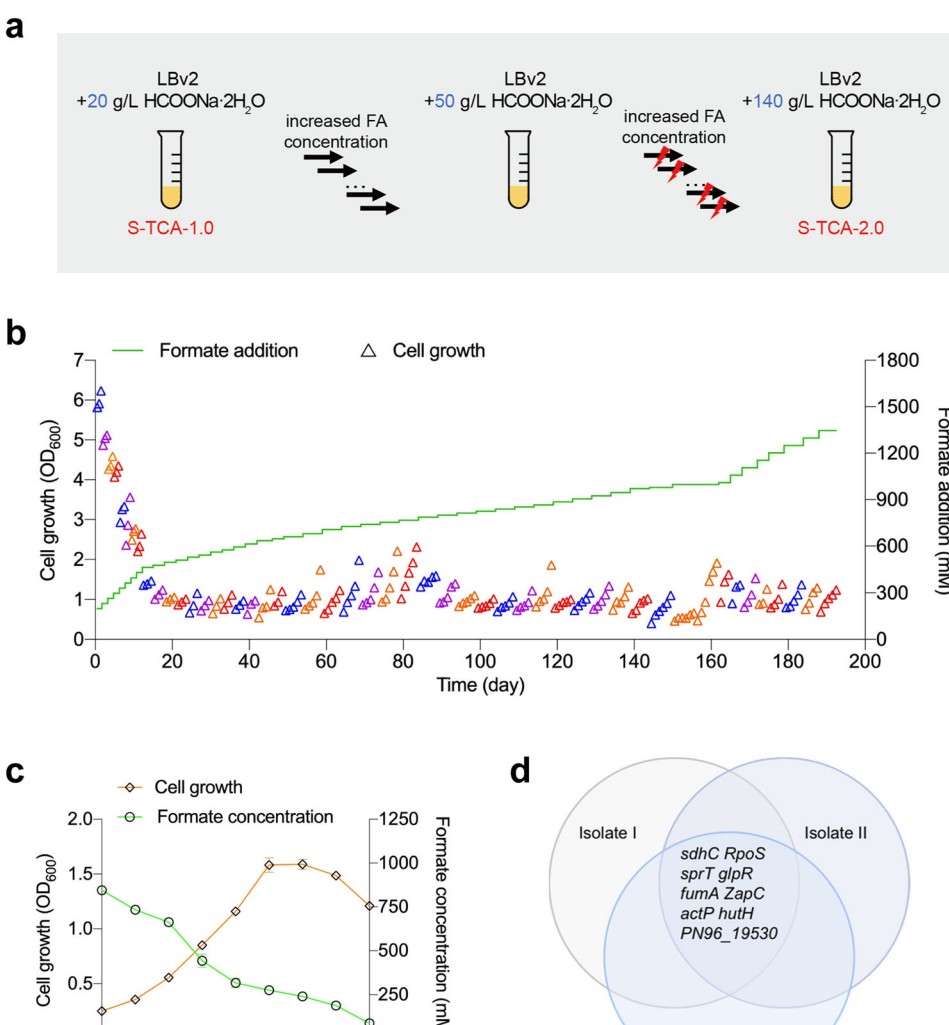

**Fig. 3 | Adaptive laboratory evolution of the *V. natriegens* strains further enhances formate tolerance and metabolism. a** Flowchart for the adaptive laboratory evolution of the *V. natriegens* S-TCA-1.0. S-TCA-2.0 was the final evolved strain with significantly enhanced formate metabolism ability in LBv2 medium. The lightning symbols represent UV mutagenesis, which was carried out every three passages, aiming to accelerate the strain mutation and evolutionary efficiency. **b** Evolution trajectory of S-TCA-1.0 to obtain the S-TCA-2.0 strain with gradually enhanced formate utilization. LBv2 medium was used with an increasing sodium formate (HCOONa·2H$_2$O) concentration from 20 to 140 g · L$^{-1}$ (192 to 1346 mM equivalent formate). Triangles indicate the bacterial biomass of each passage; the same colors represent passages with the same formate concentration. **c** Validation of the growth and formate consumption of S-TCA-2.0 within 24 h using the LBv2 medium initially supplemented with 85 g · L$^{-1}$ sodium formate (817 mM equivalent formate). Data are presented as the mean ± SD (*n* = 3 biologically independent samples). **d** The nine mutated genes occurred in all the three isolates of the S-TCA-2.0 stain. Source data are provided as a Source Data file.

respectively, are commonly observed to be mutated in ALE experiments[38]. Thus, the mutations of these three genes are unlikely to be specifically linked to the phenotypes (high formate tolerance and consumption) of S-TCA-2.0. Of note, we found a mutation (A44E) in *fumA*, the gene encoding fumarate hydratase that converts fumarate toward malate (a main flux branch point of the S-TCA cycle). The mutation of this enzyme may reduce the diversion of metabolic flux from the S-TCA cycle. Another potentially key mutation is in *sdhC*, encoding a cytochrome subunit of succinate dehydrogenase (catalyzing the conversion of succinate to fumarate in the TCA cycle), whose inactivation may change the metabolic flux of TCA cycle. For the remaining four mutated genes, there is currently no clue linked to formate consumption.

Next, we sought to evaluate how much formate can be assimilated in S-TCA-2.0. For this we cultivated this strain in the M9 minimal medium supplemented with $^{13}$C-labeled sodium formate and then collected cells for isotopomer analysis. Based on the specific

consumption rates of formate, glucose, and glycine, the specific formate assimilation rate was calculated to be 43.3 mg·gDCW$^{-1}$·h$^{-1}$, accounting for approximate 12.1% of the total consumed formate (Supplementary Table 2). Considering that some formate could be converted to the metabolites that are secreted outside cells, the exact amount of formate assimilated by this strain should be higher.

## Engineering S-TCA-2.0 for efficient synthesis of indigoidine from formate

Considering the excellent formate utilization ability of S-TCA-2.0, we attempted to evaluate if it could be reprogrammed to efficiently convert formate into valuable compounds. Here, indigoidine was chosen as a target product because it is an important pigment widely used in food, medicine, and dyeing industries, and moreover, can be synthesized using α-ketoglutarate in the TCA cycle as the precursor.

The indigoidine-producing pathway consisting of the *Streptomyces lividans* Idgs and *Bacillus subtilis* Sfp, which catalyze the

**Table 2 | The nine mutated genes occurred in all the three isolates of the S-TCA-2.0 stain**

| Name | Gene ID | Gene product | Mutation type |
|------|---------|--------------|---------------|
| RpoS | PN96_01115 | RNA polymerase sigma factor RpoS | SNP |
| sprT | PN96_00870 | SprT family zinc-dependent metalloprotease | SNP |
| ZapC | PN96_05610 | Cell division protein ZapC | SNP |
| actP | PN96_14245 | Cation acetate symporter | SNP |
| glpR | PN96_01950 | DeoR/GlpR family transcriptional regulator | SNP |
| fumA | PN96_04250 | Fumarate hydratase | SNP |
| hutH | PN96_17630 | Histidine ammonia-lyase | SNP |
| sdhC | PN96_09300 | Succinate dehydrogenase cytochrome b556 subunit | Indel |
| PN96_19530 | PN96_19530 | ATP-binding protein | SNP |

conversion of glutamine to indigoidine, was introduced into S-TCA-2.0 via an expression plasmid, yielding the strain S-TCA-2.0-IE (Fig. 4a). Next, the S-TCA-2.0-IE strain was cultivated in the LBv2 medium supplemented with $85 \, g·L^{-1}$ sodium formate ($HCOONa·2H_2O$). As expected, the culture of S-TCA-2.0-IE appeared blue color, which was caused by indigoidine, after 24 h of fermentation, whereas no blue occurred for the control (the strain S-TCA-2.0 harboring a blank plasmid) (Fig. 4b), indicating the production of indigoidine by S-TCA-2.0-IE in the presence of high concentration of formate. To further validate the formate assimilation in S-TCA-2.0-IE, the amino acids formed from the artificial S-TCA cycle (glycine, serine, isoleucine, lysine, methionine, aspartic acid, glutamate, and phenylalanine) or pyruvate (alanine, valine, and leucine) were analyzed through the isotopic labeling experiment using $^{13}C$-labelled formate. The result showed that all the detected amino acids were labeled by $^{13}C$ (mostly, $m + 1$), in which serine and methionine contained higher proportions of $^{13}C$-labelled amino acids (45.7% and 39.4%, respectively), probably due to that these two amino acids relative to the other amino acids are more closely linked to formate assimilation (THF cycle) (Fig. 4c).

We next attempted to explore the real potential of S-TCA-2.0-IE in using formate for indigoidine production in a fed-batch mode. This strain was grown in the LBv2 medium containing approximately $60 \, g·L^{-1}$ initial sodium formate ($HCOONa·2H_2O$) (equivalent to 577 mM formate), with additional sodium formate ($60 \, g·L^{-1}$) added into the medium every 12 h (repeated five times). We observed that S-TCA-2.0-IE continued to grow with the supplementation of formate, achieving the highest biomass ($OD_{600}$ ~ 3.1) at 60 h (Fig. 4d, left); in contrast, the biomass of S-TCA-2.0-IE stopped to increase after 18 h in the medium without formate addition, reaching to the highest biomass of $OD_{600}$ ~ 1.5 (Fig. 4d, left). Concurrently, S-TCA-2.0-IE consumed all the formate (approximately $165.3 \, g·L^{-1}$ formate) added into the medium and produced $29.0 \, g · L^{-1}$ of indigoidine within 72 h (Fig. 4d, right), achieving a specific formate consumption rate of $2.3 \, g · L^{-1} · h^{-1}$. Such an indigoidine production level, to our knowledge, is higher than the reported highest indigoidine production through microbial synthesis although under different cultivation conditions (Supplementary Table 3). In contrast, the indigoidine produced by S-TCA-2.0-IE was only $4.7 \, g · L^{-1}$ without the supplementation of formate (Fig. 4d, right). Altogether, these results confirmed the great potential of S-TCA-2.0-IE in indigoidine production using formate as the co-substrate.

## Discussion

FA chemically converted from $CO_2$ is an attractive feedstock for biological production of various value-added chemicals. In this study, we have two important findings. First, the exceptional tolerance and

metabolic capacity of *V. natriegens* to formate represent a superior FA-utilizing platform strain. Second, a combination of rational metabolic remodeling that generated the non-native S-TCA loop and laboratory evolution represents a successful strategy with potential universality for enhancing microbial FA utilization.

A large number of pathways and genes were found to be active during the formate metabolism in *V. natriegens*. Therefore, it seems that this organism has metabolic flexibility and versatility in the presence of formate. Multiple formate dehydrogenase genes were significantly up-regulated under formate stress (Fig. 2a), indicating that these FDHs were activated to oxidize the available formate to $CO_2$ with the generation of reducing power. Although the key genes (*ftl* and *folD*) responsible for the THF cycle exhibited similar expression levels with and without formate (Fig. 2a), the majority of the genes associated with the downstream reductive glycine pathway, the serine cycle, and the TCA cycle were significantly up-regulated following formate addition (Fig. 2a), which would pull the flux of the THF cycle and promote formate conversion into various metabolites. Noticeably, *V. natriegens* still remained partial formate consumption ability even after the deletion of all the predicted *fdh*, *ftl*, and *pfl* genes (Fig. 2b, bottom), suggesting the existence of potential alternative pathways for formate assimilation. One possibility is the pathways starting with the reduction of fomate to formaldehyde[4]: (i) formate can be ligated with CoA by an acetyl-CoA synthetase to form formyl-CoA followed by the reduction to formaldehyde by an acyl-CoA reductase. (ii) formate can be activated by a kinase to produce formyl phosphate followed by the reduction to formaldehyde by a formyl phosphate reductase. These two pathways have been constructed and tested in vivo recently[39–41]; however, whether they occur in nature in microorganisms and contribute to formic assimilation remain unclear.

Formate assimilation has been previously used to relieve the serine-auxotrophy of *E. coli*[2,42], suggesting that formate assimilation fluxes could be directed toward serine synthesis. In our design, serine production from glycolysis was blocked by deleting the *serA* gene, which would enhance the THF cycle flux to form serine. Additionally, the artificial S-TCA cycle created by integrating the serine cycle and the TCA cycle enabled the high-flux TCA to pull the upstream carbon supply and consequently enhance formate assimilation and conversion. This metabolic design is crucial for the following rapid evolution of *V. natriegens* into the strains with exceptional formate metabolic capacity since the wild-type strain did not achieve the goal using the same adaptive laboratory evolution strategy. Although the growth of *V. natriegens* was impaired after multiple gene deletions for constructing the S-TCA module (Fig. 2d), the following adaptive laboratory evolution rapidly recovered cellular fitness. This finding indicates that *V. natriegens* has excellent plasticity in metabolic remodeling. Therefore, the rapid generation of the evolved strain S-TCA-2.0 with strikingly enhanced formate metabolism should be attributed to the combination of metabolic engineering and adaptive laboratory evolution. This metabolic design may also work in other microbial chassis to create diverse cell factories for formate utilization. However, even though the S-TCA-2.0 stain exhibited efficient formate consumption, potential bottlenecks may still exist in the artificial S-TCA cycle. For example, we found that the overexpression of some genes in the glycine cleavage system (GCS) and the serine cycle could affect the formate consumption of S-TCA-2.0 (Supplementary Fig. 10), indicating ample room for further improvement of this metabolic pathway.

It should be noted that formate assimilation to acetyl-CoA via the THF cycle in microorganisms is an energy consuming process. The energy required for formate assimilation should depend on other approaches, such as the metabolism of other organic carbon sources or FDH-catalysed formate oxidation. Moreover, all the reported native and artificial bacteria showed poor growth using formate as the sole organic carbon source[5–7,9,10,19,20]. Therefore, the LBv2 medium which contains yeast extract and peptone was used here to evaluate the real

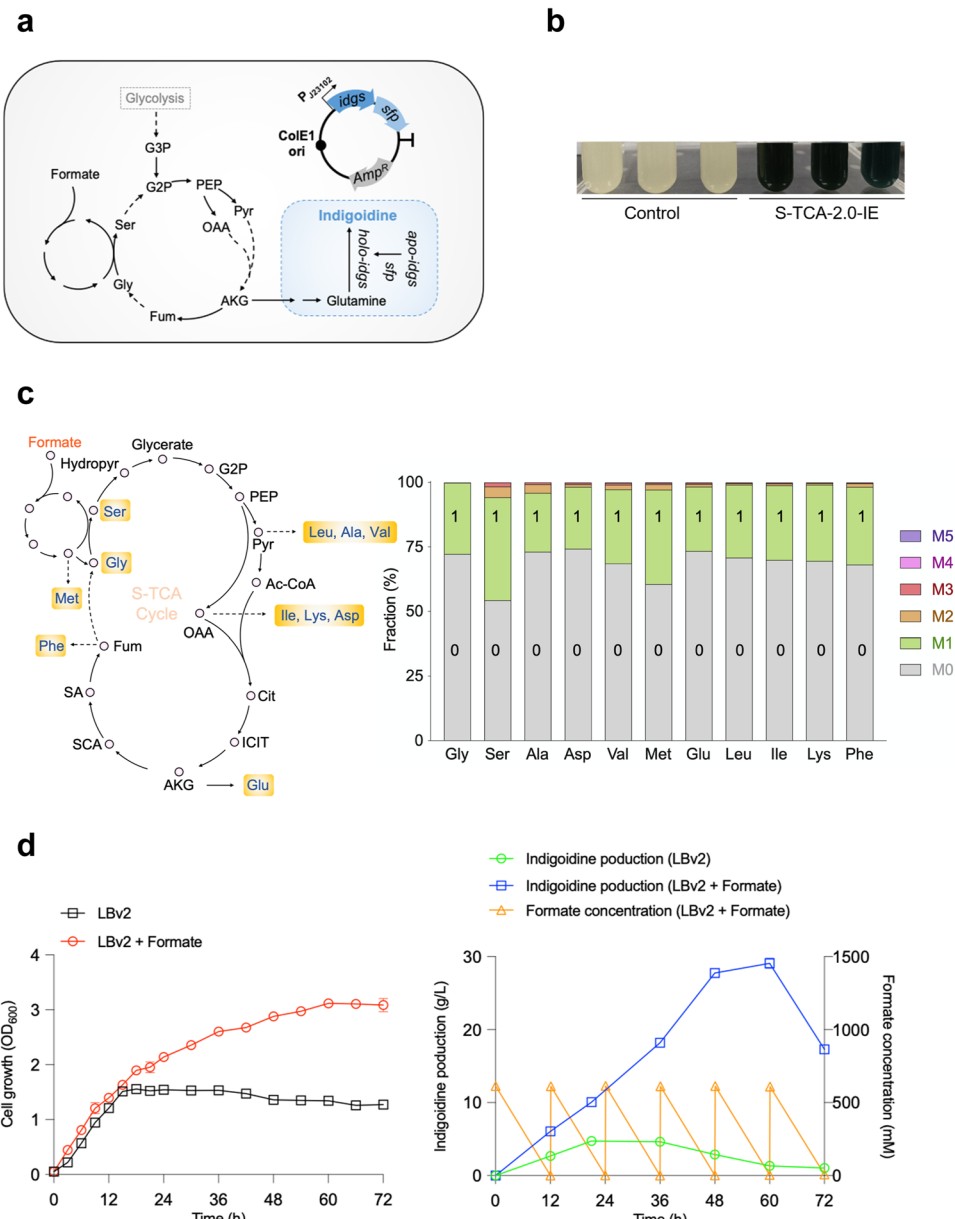

**Fig. 4 | Engineering the evolved *V. natriegens* strain for efficient production of indigoidine from formate. a** The synthetic pathway and plasmid for indigoidine production. Idgs: indigoidine synthetase. Sfp: phosphoryltransferase. **b** Indigoidine production by S-TCA-2.0-IE. The S-TCA-2.0 strain harboring a blank shuttle vector was used as the control. Strains were grown in LBv2 medium for 24 h, with initial supplementation of 85 g · L⁻¹ sodium formate (HCOONa·2H₂O). **c** The average ¹³C fraction of eleven analyzed amino acids of the S-TCA-2.0-IE strain grown in the LBv2 media supplemented with both unlabeled (40 g·L⁻¹) and ¹³C-labeled (60 g·L⁻¹) sodium formate (HCOONa·2H₂O). The left panel represented the metabolic pathways for the formation of amino acids, in which dashed lines represent the multiple-step pathways. Data are presented as the mean ± SD ($n = 3$ biologically independent samples). **d** Growth (left), indigoidine production (right), and formate consumption (right) of S-TCA-2.0-IE in a fed-batch mode. Initial sodium formate (HCOONa·2H₂O) concentration in medium was 60 g·L⁻¹. Extra sodium formate (60 g·L⁻¹) was added into the medium every 12 h for five times. Hydropyr hydroxypyruvate, PEP phosphoenolpyruvate, OAA oxaloacetate, AKG alpha-ketoglutaric acid, Cit citrate, ICIT isocitrate, SA succinic acid, SCA Succinyl-CoA, Pyr pyruvate, G2P 2-Phospho-D-glycerat, G3P 3-Phospho-D-glycerate, Mal malate, Fum fumarate, Gly glycine, Ser serine, Met methionine, Leu leucine, Ala alanine, Val valine, Ile isoleucine, Asp aspartic acid, Glu glutamic acid, Phe phenylalanine, Lys lysine. Data are presented as the mean ± SD ($n = 3$ biologically independent samples). Source data are provided as a Source Data file.

potential of the strain S-TCA-2.0-IE in formate consumption and indigoidine production, rather than using the minimal medium containing formate as the sole carbon source. Although yeast extract and peptone in the LBv2 medium enabled growth and indigoidine synthesis of S-TCA-2.0-IE, further enhanced biomass and indigoidine production were observed with the supplementation of formate (Fig. 4d), strongly indicating the contribution of formate consumption to indigoidine production. We analyzed the proportion of ¹³C-labelled glutamine (the precursor of indigoidine) to total glutamine in the S-TCA-2.0-IE strain and found that the ¹³C-glutamine (M1) accounted for 12.9% of total

glutamine (Supplementary Fig. 11). This analysis confirmed that the carbon atom in formate could be incorporated into indigoidine but the ratio was not high. Therefore, we deduce that a large portion of the consumed formate was oxidized to form $CO_2$ (generating reducing power) and assimilated into biomass components in S-TCA-2.0-IE, which would facilitate cell growth or other co-substrates (e.g., tryptone and yeast extract) to be more used for indigoidine synthesis. This may also partially explain why the addition of formate could significantly enhance the production of indigoidine by S-TCA-2.0-IE grown in the LBv2 medium.

In summary, our work uncovered the excellent metabolic power of *V. natriegens* towards formate and further developed this bacterium into an efficient cell factory for converting formate into target products. The knowledge gained in this study and the engineering strategies presented here will be highly useful for advancing formate bioutilization.

## Methods

### Bacterial strains and growth conditions

*Vibrio natriegens* ATCC 14048 (stored at −80 °C as a 20% glycerol stock) was used as the parental strain for formate fermentation as well as further genetic modifications. The *V. natriegens* strains were grown in the nutrient-rich LBv2 medium or a slightly modified M9 minimal medium (addition of 2% (w/v) NaCl; 0.25 g·L$^{-1}$ of glycine was added when needed) at 30 °C[22]. Chloramphenicol (12.5 µg·mL$^{-1}$), kanamycin (100 µg·mL$^{-1}$), carbenicillin (100 µg·mL$^{-1}$), and different amounts of sodium formate (HCOONa·2H$_2$O) were added into the media when needed. *Escherichia coli* DH5α (stored at −20 °C as a 20% glycerol stock) was used as the host for plasmid cloning. The *E. coli* cells were grown in the LB medium supplemented with chloramphenicol (12.5 µg·mL$^{-1}$), ampicillin (100 µg·mL$^{-1}$), or kanamycin (100 µg·mL$^{-1}$). All the strains used in this study are listed in Supplementary Table 4.

### Reagents and chemicals

Primers were synthesized by BioSune (BioSune, China). PCR reactions were carried out by using KOD plus Neo and KOD FX DNA polymerase (Toyobo, Japan). The restriction enzymes and ligases used in plasmid construction were purchased from Thermo Fisher Scientific (Thermo Fisher Scientific, USA) and Takara (Takara, Japan), respectively. The assembly of multiple DNA fragments for plasmid construction was performed by using the ClonExpress MultiS One Step Cloning Kit (Vazyme Biotech Co., Ltd., China). Plasmid isolation and DNA purification were performed with kits (Axygen Biotechnology Company Limited, China).

Sodium formate used in this study was HCOONa·2H$_2$O. $^{13}$C-labelled sodium formate was ordered from Cambridge Isotope Laboratories (Cambridge Isotope Laboratories, Inc., USA). The other commercial chemicals were purchased from Sigma-Aldrich (Sigma-Aldrich Co., USA) and Aladdin (Shanghai Aladdin Biochemical Technology Co., Ltd., China).

### Plasmid construction

All the plasmids and primers used in this study are listed in Supplementary Table 4 and Supplementary Data 1, respectively.

The vector co-expressing *idgs* (SLIVYQS_19313, encoding indigoidine synthetase) and *sfp* (GFX43_019515, encoding phosphoryltransferase), from *S. lavendulae* CGMCC 4.1386 and *Bacillus subtilis* 168, respectively, for indigoidine production was constructed as follows. A DNA fragment (*idgs*-RBS-*sfp*) that contained the codon-optimized *idgs* gene, an RBS sequence (ggatccactctgtcagatctcactctgccaaggaggacgcac), and the *sfp* gene was synthesized (GenScript, China). Another DNA fragment that harbored a promoter (ttgacagctagctcagtcctaggtactgtgctagc) and RBS sequence (agagtcacacaggaaagtacta) was obtained by PCR amplification, using the primers J23102-B00320m-F/J23102-B00320m-R which have a 21-nt overlap at the 3′ region. Next, this DNA fragment was linked to the *idgs*-RBS-*sfp* fragment via overlapping PCR using the primers J23102-B00320m-idgs-sfp-F/idgs-sfp-R, yielding a larger DNA fragment. The obtained DNA fragment was further linked with a linear plasmid pColE1-Amp vector[43] using a ClonExpress MultiS One Step cloning kit (Vazyme, China), generating the target plasmid for indigoidine production.

The plasmid pMMB67EH-tfox-sacB, modified from the plasmid pMMB67EH-tfox[28], was used for supporting the transformation of exogenous DNA into *V. natriegens*. In brief, the plasmid backbone was obtained by PCR amplification using the plasmid pMMB67EH-tfox as the template and the primers tfox-F/tfox-R. The *sacB* gene was obtained by PCR amplification using the plasmid pEcCas[44] as the template and primers sacB-F/sacB-R. These two DNA fragments were assembled using the ClonExpress MultiS One Step cloning kit (Vazyme, China), yielding the plasmid pMMB67EH-tfox-sacB.

### Generation of *V. natriegens* mutants

Gene knockout was achieved in *V. natriegens* using the MuGENT (multiplex genome editing by natural transformation) method[28]. In brief, the *V. natriegens* cells harboring the abovementioned pMMB67EH-tfox-sacB plasmid were grown overnight in the LBv2 medium (containing 100 µg·mL$^{-1}$ carbenicillin and 100 µM IPTG) in a shaking incubator with a shaker rate of 280 rpm at 30 °C. 3.5 µL of the grown culture were then diluted into 350 µL using the IO medium (28 g·L$^{-1}$ of instant ocean sea salts)[28] plus 100 µM IPTG, which was further mixed with 50 ng of a selected product (a DNA fragment that has an antibiotic resistance marker and two homologous arms flanking the *dns* gene) and 200 ng of an unselected product (a DNA fragment that has two homologous arms flanking the gene of interest but lacks any selectable marker). The mixture was incubated statically at 30 °C for 5 h. Next, 1 mL of the LBv2 medium was added, and the mixture was further outgrown at 30 °C with shaking (280 rpm) for 2 h. The cultures were then plated onto the LBv2 agar plates (containing 100 µg·mL$^{-1}$ carbenicillin and 12.5 µg·mL$^{-1}$ chloramphenicol or 100 µg·mL$^{-1}$ kanamycin). When colonies were visible on the plates after incubation at 30 °C for 10 h, some were picked and subjected to colony PCR using the primers KO-gene-seq-F/KO-gene-seq-R, aiming to detect the desired gene knockout events.

Finally, to eliminate the pMMB67EH-tfox-sacB plasmid in cells, the colonies with desired gene knockout were harvested from agar surface, resuspended in 5 mL of the LBv2 medium (containing 80 g·L$^{-1}$ sucrose), and incubated at 30 °C with shaking (280 rpm) for 10 h. The grown cells were plated onto the LBv2 agar plates (containing 80 g·L$^{-1}$ sucrose) again. When colonies were visible on the plates after incubation at 30 °C for 12 h, some were picked and resuspended in the LBv2 medium with and without carbenicillin (100 µg·mL$^{-1}$) to verify plasmid curing.

### Adaptive laboratory evolution

The engineered *V. natriegens* strain S-TCA-1.0 was first inoculated into 5 mL of the LBv2 medium that contained 20 g·L$^{-1}$ sodium formate (HCOONa·2H$_2$O) and then incubated in a shaking incubator (30 °C and 280 rpm) for inoculum preparation. After overnight cultivation, 0.1 mL of grown cells were inoculated into 5 mL of the fresh LBv2 medium (containing 20 g·L$^{-1}$ sodium formate). Once cells could grow to over 1.0 of OD$_{600}$ after overnight cultivation by serial subculturing, the concentration of sodium formate was increased in the next round of subculturing. Samples were taken every passage and then stored at −80 °C. When formate concentration in the medium was raised to 50 g·L$^{-1}$, the UV (ultraviolet) mutagenesis of cells was carried out every three passages by placing the culture at a distance of 10 cm away from a 20-W UV light lamp for 15 min, aiming to accelerate the strain mutation and evolutionary efficiency.

### Analytical methods

The concentration of formate was determined using an Agilent 1100 HPLC system (Agilent, China) equipped with a bio-Rad Aminex HPx-87H column (Bio-Red). In brief, samples were taken from the fermentation broth at appropriate time intervals and then centrifuged at 7000 × g for 20 min at 4 °C. Next, 200 µL of supernatant was analyzed by HPLC. The column temperature was maintained at 30 °C. The flow rate of the mobile phase (5 mM H$_2$SO$_4$) is 0.6 mL·min$^{-1}$.

The purification of indigoidine was performed as previously described with slight modifications[45]. In brief, the fermentation broth was firstly freeze-dried. The generated powder was sequentially

treated with methanol, isopropanol, water, ethanol, and hexane for three rounds, aiming to lyse cells and remove contaminating proteins and metabolites. The pellet was dried overnight and then resuspended in dimethylsulfoxide (DMSO) at a concentration of $1 \, mg \cdot mL^{-1}$. The purity of indigoidine was determined by HPLC and mass spectrum (Supplementary Fig. 12). The purified indigoidine reagent was used to establish a standard curve correlating the concentration of indigoidine to the absorbance at $OD_{600}$ (Supplementary Fig. 13). To quantify indigoidine produced by *V. natriegens*, 1 mL of the fermentation broth was collected and treated by the same procedure described above. The concentration of indigoidine was determined by measuring the optical absorbance at 600 nm using a spectrophotometer (DU730, Beckman Coulter, Inc., USA).

The measurement of glycine was carried out based on the method described previously with a few modifications[46]. In brief, glycine was measured by HPLC with o-phthaldialdehyde (OPA) (500 mg OPA, 5 ml ethanol, and 500 μl mercaptopropionic acid in 50 ml of 0.4 M borate buffer) as a precolumn derivatization reagent. Mobile phase A and B were sodium acetate solution (50 mM) and methanol, respectively. The flow rate was $1 \, mL \cdot min^{-1}$ and the detection wavelength was 334 nm. The Zorbax Eclipse XDB-C8 (4.6 × 150 mm, 3.5 μm) column was used with a matched guard column. The gradient program for HPLC was as follows: 0–7 min, 30% solution B; 7–14 min, 45% solution B; 14.5–18 min, 30% solution B. The sample loading volume was 5 μl. The column temperature was kept at 35 °C.

## RNA-seq analysis

*V. natriegens* strains were cultured in the LBv2 medium with or without the supplementation of sodium formate ($HCOONa \cdot 2H_2O$) ($40 \, g \cdot L^{-1}$). The grown cells (6 h of cultivation) were harvested by centrifugation ($6000 \times g$, 4 °C, 20 min) and then frozen in liquid nitrogen. The RNA-seq assays were finished by the Majorbio Co. Ltd. in China. In brief, total RNA was extracted using the TRIzol® reagent (R0016, Beyotime, China) followed by the removal of genomic DNA using DNase I (TaKaRa, Japan). The 16S and 23S ribosomal RNA (rRNA) in total RNA were eliminated using a Ribo-Zero Magnetic kit (Epicenter Bio-technologies, WI, USA). Thereafter, mRNA was broken into short fragments (~200 bp) with the fragmentation buffer. Next, cDNA was synthesized via reverse transcriptase using a SuperScript double-stranded cDNA synthesis kit (Invitrogen, CA) with random hexamer primers (Illumina, USA). The second strand of cDNA was generated by incorporating deoxyuridine triphosphate (dUTP) into the place of deoxythymidine triphosphate (dTTP), aiming to create blunt-ended cDNAs. The yielding double-stranded cDNAs were subjected to end repair, phosphorylation, 3' adenylation, and adapter ligation in sequential. Next, the second strand of cDNA with dUTP was degraded using UNG enzyme (Uracil-N-Glycosylase). cDNA fragments were separated on a 2% agarose gel. The DNA fragments of ~200 bp were extracted from the gel and used as the template for the construction of cDNA libraries by PCR amplification using the Phusion DNA polymerase (NEB, USA) with 15 reaction cycles. After quantification with a micro fluorometer (TBS-380, TurnerBioSystems, USA), the libraries were sequenced on the Illumina HiSeq × Ten sequencer using paired-end sequencing.

Bioinformatics analysis was performed using a cloud platform (Shanghai Majorbio Bio-Pharm Technology Co., Ltd., China) based on the data generated by the Illumina platform. The main methods are as follows: RSEM was used to quantify gene and isoform abundances. The TPM method was used to calculated expression levels. Differentially expressed genes were identified by using the DESeq2 packages (http://bioconductor.org/packages/release/bioc/html/DESeq2.html).

## Quantitative real-time-PCR

Quantitative real-time-PCR (qRT-PCR) was performed to verify the RNA-seq data. Briefly, cultures (cultivation in the LBv2 medium with or without the supplementation of sodium formate) were taken after 6 h of cultivation, frozen quickly in liquid nitrogen, and ground into powders. RNA samples were then extracted using a Ultrapure RNA Kit (SparkJade, China), followed by the removal of genomic DNA with DNase I (SparkJade, China). Next, qRT-PCR was carried out in a MyiQ2 thermal cycler (Bio-Rad, USA) using a SYBR Green PCR premix kit (SparkJade, China). The transcript levels of the tested genes were normalized to that of the 16S rRNA gene (PN96_00205, an internal control) of *V. natriegens*. The primers used are listed in Supplementary Data 1.

## $^{13}C$ isotopomer analysis

$^{13}C$ isotopomer analysis of amino acids in the *V. natriegens* S-TCA-2.0-IE strain was carried out as follows. Briefly, the *V. natriegens* cells were grown in 4 mL of the LBv2 medium ($40 \, g \cdot L^{-1}$ of $HCOONa \cdot 2H_2O$) at 30 °C. When cells grew to the plateau stage, $60 \, g \cdot L^{-1}$ of $^{13}C$-labeled sodium formate ($HCOONa \cdot 2H_2O$) was added into the medium. After 24 h of cultivation, 0.01 mL of the grown cells were inoculated into the same medium ($40 \, g \cdot L^{-1}$ of $HCOONa \cdot 2H_2O$) for cultivation again with the supplementation of $60 \, g \cdot L^{-1}$ of $^{13}C$-labeled sodium formate. Such a subculturing was repeated three times. The grown cells of the last passage were harvested by centrifugation ($6000 \times g$, 4 °C, 20 min) and incubated with 200 μL 6 M HCl at 105 °C for 24 h. The samples were dried in a vacuum centrifuge and then derivatized with 100 μL of pyridine and 50 μL of N-methyl-N-[tert-butyldimethylsilyl] tri-fluoroacetamide at 85 °C for 1 h. After filtration (0.45 μm pore size, Millipore), 1 μL of samples were injected into a GC-MS system equipped with a DB-5HT column (30 × 0.25 mm, 0.1 μm), which was operated in an electron shock (EI) mode at 70 eV. Amino acids were determined by matching masses and retention times to authenticated standards library.

Isotopomer analysis of glutamine in the S-TCA-2.0-IE strain was performed as follows. The cells were grown in 5 mL of the LBv2 medium supplemented with $60 \, g \cdot L^{-1}$ of $^{13}C$-labelled sodium formate ($HCOONa \cdot 2H_2O$). After 24 h of cultivation, 0.25 mL of the grown cells were inoculated into the same medium for cultivation again. Such a subculturing was repeated three times. The grown cells of the last passage were collected by centrifugation ($6000 \times g$, 4 °C, 20 min). Then, cells were mixed with acetonitrile/methanol/water (40: 40: 20), incubated at −20 °C for 20 min, and centrifuged ($12,000 \times g$, 4 °C, 20 min). The supernatant was collected to analyze glutamine by ultrahigh-performance liquid chromatograph (UHPLC; Acquity; Waters) coupled to a Q Exactive hybrid quadrupole-orbitrap mass spectrometer (Thermo Fisher Scientific, Inc., USA). Metabolites were separated with a Waters Corp Acquity UPLC HSS T3 column (100 × 2 mm, 1.8 μm). The mass spectrometer was run in both electrospray ionization positive (ESI+) and negative (ESI−) modes. Data were acquired using full scan over 70–1000 $m/z$ at 70,000 resolution. MS/MS spectra were acquired with 30-eV collision energy. Data processing was performed by Compound Discoverer 2.0 (Thermo Fisher Scientific, Inc., USA), with a mass tolerance of 5 ppm for both noise filtering and database search.

## Metabolomics analysis

The grown cells ($OD_{600}$ ~ 1.0) were harvested by centrifugation ($6000 \times g$, 4 °C, 20 min). Then, cells were mixed with acetonitrile/methanol/water (40:40:20) and incubated at −20 °C for 20 min. Next, the samples were centrifuged ($12,000 \times g$, 4 °C, 20 min), and the supernatant was collected. The metabolites were analyzed through the ultrahigh performance liquid chromatograph (Acquity, Waters) with a Q Exactive hybrid quadrupole−orbitrap mass spectrometer (Thermo Fisher Scientific, Inc., USA). 10 μL of the samples were injected into the LC-MS system. Metabolites were separated by a Luna NH2 column (100 mm × 2 mm, 3 μm particle size, Phenomenex) with a solvent flow rate of 0.3 mL min$^{-1}$. The analysis was carried out with the column

temperature maintained at 15 °C. Solvent A (20 mM ammonium acetate, adjusted to pH 9.0 with ammonium hydroxide) and solvent B (acetonitrile) served as the mobile phases with a gradient as follows: 0 min, 85% solvent B; 10 min, 45% solvent B; 15 min, 2% solvent B; 18 min, 2% solvent B; 18.1 min, 85% solvent B; 24 min 85% solvent B. The mass spectrometer with a heated electrospray ionization source was operated in a positive mode. Data were generated through full scan over 70–1000 $m/z$ at 70,000 resolution. MS/MS spectra were acquired with 30-eV collision energy. The compound identities were verified by mass and retention-time match to authenticated standards.

### Analysis of $^{13}$C ratio in biomass

Formate assimilation ratio was performed by analyzing the proportion of $^{13}$C in the total carbon of biomass as previously described[18]. In brief, the *V. natriegens* S-TCA-2.0 cells were grown in the modified M9 medium (supplemented with 2.0 g·L$^{-1}$ glucose, 0.25 g·L$^{-1}$ glycine, and 10 g·L$^{-1}$ $^{13}$C-labeled HCOONa·2H$_2$O) at 30 °C. After 48 h of cultivation, 1 mL of the grown cells were harvested by centrifugation (6000 × *g*, 4 °C, for 10 min) and then washed twice with the double distilled water. Next, the samples were freeze-dried and used for determining $^{13}$C contents by an isotope ratio mass spectrometer coupled with an elemental analyzer (Thermo Fisher Scientific, Inc., USA).

### Reporting summary

Further information on research design is available in the Nature Portfolio Reporting Summary linked to this article.

## Data availability

The RNA-seq data and genome resequencing data generated in this study have been deposited in the NCBI SRA database under Bioproject accession PRJNA843433 and PRJNA1022572, respectively. Source data are provided with this paper.

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

## Acknowledgements

This research was supported by the National Key R&D Program of China (No. 2018YFA0901500 to Y.G.), the National Natural Science Foundation of China (No. 31921006 to W.J.), the Shanghai Science and Technology Commission (No. 21DZ1209100 to Y.G.), DNL Cooperation Fund, CAS (No. DNL202013 to Y.G.), and Tianjin Synthetic Biotechnology Innovation Capacity Improvement Project (TSBICIP-KJGG-016-01 to Y.G.). We are grateful to Wenli Hu and Xiaoyan Xu for HPLC and LC-MS/MS analyses.

## Author contributions

J.T. conceived and performed strain modifications, adaptive laboratory evolution and isotopic labeling experiments; W.D. and Z.Z. constructed plasmids; J.T. and W.D. analyzed the phenotypic changes of the mutants; W.D. performed bioinformatics analyses and isotopic labeling experiments; Y.G. and W.J. supervised and directed the study. Y.G., W.J., J.T. and W.D. wrote the manuscript. G.Y., G.Z., S.Y., and J.X. discussed results and offered advices.

## Competing interests

The authors declare no competing interests.
