## [Peer Review File · Nature Communications]

Discovery and remodeling of *Vibrio natriegens* as a microbial platform for efficient formic acid biorefineryReviewers' Comments:

Reviewer #1:

Remarks to the Author:

Formic acid (FA), a one-carbon source derivable from CO₂, is seen as a promising feedstock for carbon biorefinery. The authors found that *Vibrio natriegens*, a gram-negative bacterium, has excellent native FA tolerance and metabolic capacity. Through bioinformatics and transcriptomic analyses, formate metabolic pathways in *V. natriegens* were identified, and engineered by creating a metabolic loop for the enhanced FA assimilation. The modified strains showed significantly improved performance using FA as a carbon source. One strain, when given a foreign indigoidine-forming pathway, achieved an exceptional FA consumption rate. In this study, the authors implemented a creative strategy of efficiently utilizing FA for the production of industrial chemicals. The reviewer has comments on the S-TCA, which is the core of this study.

1. Page 9, Lines 127 and 128: Please provide the authors' insights on potential alternative pathways for formic acid assimilation.

2. Pages 9, Lines 131 and 135: Integrating the TCA cycle and the serine cycle is a creative strategy, but it requires more explanation.

i) Is the observed decrease in cellular growth within an acceptable range? Because the cellular growth is an important parameter with respect to the formate consumption, please provide convincing explanation.

ii) Please provide evidences that the S-TCA truly contributed to the enhanced formate consumption. All the S-TCA genes need to be properly expressed, and the metabolites throughout the S-TCA should have certain level of concentrations.

Reviewer #2:

Remarks to the Author:

Comments on manuscript NCOMMS-23-21436-T by Tian et al.

Major Comments

- The authors used sodium formate in this study and compared the effects on growth and transcription of *V. natriegens* under varying concentrations. Sodium is essentially required for growth of this organism but higher concentrations have negative effects on growth (Hoffart et al. <https://doi.org/10.1128/AEM.01614-17>). Therefore, it is not clear whether the observed effects derive from sodium, formate or both. Please perform control experiments for growth in complex LBv2 medium and minimal medium where you apply the respective sodium concentrations without formate. Since sodium probably also plays an important role in energy conservation and transport mechanisms in this organism these experiments are essentially required. Please also apply such control experiments for the transcriptomic analysis. Of course, you can also use formic acid throughout the work. As stated in the introduction formic acid would be the substrate of choice. This is an important issue since the utilization of sodium formate creates a huge salt load that has to be handled on an industrial scale.

- In Fig. 1 a and b growth experiments are performed in LBv2 complex medium. In Fig. 2 b growth in minimal medium is shown, however, with much lower formate concentrations compared to the experiments with LBv2. Please perform a set of experiments in minimal medium with varying formate concentrations. Probably, the claimed (l. 71 f) high tolerance is less pronounced in minimal medium. If so, this should be carefully described. Moreover, it would be interesting to see if in minimal medium with glucose the addition of formate also increases the biomass yield.

- The authors applied ALE to improve the formate consumption of the engineered strain S-TCA-1.0. Please re-sequence the evolved strain S-TCA-2.0 and provide the mutations in the genome. This

information is also essential to understand the mechanism of formate assimilation. For the ALE approach, the question arises whether the strain is evolved to tolerate high sodium or formate concentrations or both.

- Since the strain with deleted *fdh*, *ftl*, and *pfl* genes still shows significant formate consumption and growth (Fig. 2 b), it is not clear how formate assimilation works in *V. natriegens*. Please provide flux analysis to get a more comprehensive picture of the formate metabolism. This is important because the phenotype of the *fdh* deletion strain is surprising (Fig. 2 b). One would assume that this reaction is essential for growth on formate.

- Indigoidine production: Please provide the maximal theoretical and achieved product yield on formate. Please consider for the calculation the requirements for reduction equivalents. After 60 h, indigoidine is degraded although formate is still present. What is the reason? Can you provide information about the production in minimal medium?

Reviewer #3:

Remarks to the Author:

Vibrio natriegens has received large attention for its favorable properties of fast growth and uses of syngas and C1 carbons. This manuscript first presents a through omics study of *V. natriegens* grown on formate. Metabolic engineering approach was applied to generate a formate sink by linking formate assimilation with a fused cycle of the serine and the TCA cycles, resulting in a strain S-TCA-1.0. Adaptive laboratory evolution of S-TCA-1.0 was done to create a high-performance strain which can grow in a medium with sodium formate as high as 140 g/L with the consumption of 78.9 g·L⁻¹ sodium formate within 24 h following initial supplementation of 85 g·L⁻¹ sodium formate in the medium. The cellular adaptation of the wild-type strain under formate stress was found to be much slower than that of S-TCA-1.0, suggesting that the reconstituted formate metabolic pathways in *V. natriegens* (S-TCA-1.0) facilitated the strain evolution to adapt to high formate stress. The strain S-TCA-2.0 was further engineered for efficient synthesis of indigoidine from glutamate first and then from formate. This work could be of significant interest and impact if the major issues as stated below could be properly addressed.

Major issues:

1. Although the authors claimed that a high formate consumption rate was achieved in the engineered *V. natriegens*, the novelty and experimental design of this work need more clarification. Fundamentally, they did all the experiments on complex medium-LB containing yeast extract, peptone and others; this makes the mechanism of formate metabolism obscure and intractable. For example, as shown in Fig. 2c+4a, glycolysis contributed to the formation of the precursors for formate assimilation but it is not clear what level of glycolytic flux is required to stimulate or maintain a formate assimilation?
2. Can the strain grow in a minimal medium with formate as the only carbon source or a co-substrate (even though the growth rate may be low)? What is the carbon yield and how much formate is assimilated into the products as well as the biomass? These information/data are very important to evaluate the real potential of *V. natriegens* as a C1 utilizer.
3. In Fig. 4c, the results of C13 labelling experiments are not clearly described. Only one sentence for this is completely not enough. "The result showed that all the detected amino acids were labeled by C13". How much formate goes into the amino acids? Are there any differences in terms of C13 composition, in different amino acids (some are derived from PEP, some from TCA intermediates)? It seems that a large fraction of amino acids is unlabeled.
4. As the authors mentioned, the genome of *V. natriegens* has more than six formate dehydrogenase genes. So it has a strong formate dissimilation capacity to degrade formate into CO₂ instead of using formate as a carbon donor. The authors highlighted several times that *V. natriegens* could metabolize a high amount of formate within 24 hs (Fig. 2e, 3c), but how much formate is oxidized to CO₂ and how much is assimilated as the carbon source? If most of the formate is converted into CO₂, what is the

significant advantage to develop such a microbial chassis (CO₂-Formate-CO₂ futile cycle)?

5. The authors should focus more on the identification of potential bottlenecks in the S-TCA cycle to stimulate formate assimilation. For example, what would be the outcome of overexpression of GCS or the serine cycle?

6. In some aspects, the transcriptomics data are hardly understandable or even misleading, e.g. *pfl* is upregulated by more than 8.9-folds, but its deletion has no effect on cell growth and formate metabolism. Pl explain it.

7. In supplementary Table 3, *V. natriegens* $\Delta fdh \Delta ftl \Delta pfl$ was mentioned to have three *fdh* genes deletion, how about the remaining three?

8. The role of formate for indigoidine formation is not clear. Formate was only shown to be incorporated into amino acids for cell growth. Carbon for indigoidine formation is from glutamate!

Point-by-point response to the referees' comments

Reviewer #1 (Remarks to the Author):

Formic acid (FA), a one-carbon source derivable from CO₂, is seen as a promising feedstock for carbon biorefinery. The authors found that *Vibrio natriegens*, a gram-negative bacterium, has excellent native FA tolerance and metabolic capacity. Through bioinformatics and transcriptomic analyses, formate metabolic pathways in *V. natriegens* were identified, and engineered by creating a metabolic loop for the enhanced FA assimilation. The modified strains showed significantly improved performance using FA as a carbon source. One strain, when given a foreign indigoidine-forming pathway, achieved an exceptional FA consumption rate. In this study, the authors implemented a creative strategy of efficiently utilizing FA for the production of industrial chemicals. The reviewer has comments on the S-TCA, which is the core of this study.

1. Page 9, Lines 127 and 128: Please provide the authors' insights on potential alternative pathways for formic acid assimilation.

Reply: Thanks for the reviewer's suggestion. The insights on the potential alternative pathways for formic acid assimilation in *V. natriegens* (as shown below) have been added into the Discussion section of the manuscript (Page 18, Line 265-273).

Based on the previous report (*Arren Bar-Even. Formate assimilation: the metabolic architecture of natural and synthetic pathways. Biochemistry, 2016, 55, 3851-3863*), we deduce that the possible alternative pathways for formic acid assimilation in *V. natriegens* are those starting with the reduction of formate to formaldehyde (as shown below, Figure 1): (i) Formate can be ligated with CoA by an acetyl-CoA synthetase to form formyl-CoA followed by the reduction to formaldehyde by an acyl-CoA reductase. This synthetic pathway has been proposed and tested in vivo (*ACS Synthetic Biology, 2022, 11, 135-143; Synthetic Biology, 2021, 6, ysab020*); **however, whether it is a natural active formate-assimilation pathway in microorganisms remains unclear.** (ii) Formate can be activated by a kinase to produce formyl phosphate followed by the reduction to formaldehyde by a formyl phosphate reductase. Such a synthetic pathway was also constructed and tested in vivo recently (*Nature Communications, 2023, 14, 2682*); **however, this pathway is also not known to occur in nature.**

Figure 1 The pathways starting with the reduction of formate to formaldehyde (*Nature Communications, 2023, 14, 2682. <https://doi.org/10.1038/s41467-023-38072-w>*). ACS, acetyl-CoA synthetase; ACR, acyl-CoA reductase; ACK, acetate kinase; FPR, formyl phosphate reductase.

In addition, we predicted three ACS-encoding genes (PN96_21120, PN96_14210, and PN96_05365) and two ACK-encoding genes (PN96_03365 and PN96_21510) in *V. natriegens* according to the genome annotations and sequence alignment (identity $\geq 30\%$) (as shown below, Table 1). Next, we deleted these genes separately in the previously constructed mutant strain that lacks *fdh*, *ftl*, and *pfl* (shown in the original manuscript) to examine whether it would cause changes in formate consumption. As shown below (Figure 2), no significant change was found after the deletion of these genes, indicating that the pathways starting with the reduction of formate to formaldehyde may not exist in *V. natriegens*. Therefore, the identification of potential alternative pathways for formic acid assimilation in *V. natriegens* require further exploration.

Table 1 The predicted ACS and ACK-encoding genes in *V. natriegens*

Reaction	Gene ID	Enzyme	Identity ^a
Formate → Formyl-CoA	PN96_14210	acetyl-CoA synthetase	73.19%
	PN96_05365	propionyl-CoA synthetase	40.42%
	PN96_21120	acetyl-CoA synthetase	35.66%
Formate → Formyl phosphate	PN96_03365	acetate kinase	74.31%
	PN96_21510	acetate kinase	52.49%

^aAmino acid sequence identity was obtained by BlastP using the reported *E. coli* acetyl-CoA synthetase gene (Gene ID: b4069) and acetate kinase gene (Gene ID: b2296) (<https://doi.org/10.1038/s41467-023-38072-w>) as the templates.

Figure 2 The influence of the deletion of PN96_14210, PN96_05365, PN96_21120, PN96_03365, and PN96_21510 on the formate consumption of *V. natriegens*. Sodium formate ($\text{HCOONa} \cdot 2\text{H}_2\text{O}$) and glucose supplemented in the M9 medium was 5.0 and 4.0 $\text{g} \cdot \text{L}^{-1}$, respectively. The control was the previously constructed mutant strain that lacks *fdh*, *ftl*, and *pfl*. ns: no significance.

2. Pages 9, Lines 131 and 135: Integrating the TCA cycle and the serine cycle is a creative strategy, but it requires more explanation.

(i) Is the observed decrease in cellular growth within an acceptable range? Because the cellular growth is an important parameter with respect to the formate consumption, please provide convincing explanation.

Reply: Thanks for the reviewer's suggestion. Although the growth of *V. natriegens* was

impaired by integrating the TCA cycle and the serine cycle, **the resulting S-TCA-1.0 strain exhibited higher formate consumption per unit biomass compared with the wild-type strain** (Fig. 2f in the manuscript), indicating an enhanced formate consumption flux in S-TCA-1.0. Thus, S-TCA-1.0 as a promising chassis was further subjected to adaptive laboratory evolution to restore its growth, yielding the S-TCA-2.0 strain.

Here, we compared the growth of the wild-type, S-TCA-1.0, and S-TCA-2.0 strains. As shown below (Figure 3), the growth of S-TCA-2.0 was recovered to the level of the wild-type strain. Therefore, we think that the integration of the TCA cycle and the serine cycle followed with adaptive laboratory evolution is a feasible strategy for developing applicable *V. natriegens* strains in formate utilization.

Figure 3 Comparison of the growth of the wild-type (WT), S-TCA-1.0, and S-TCA-2.0 strains. The LBv2 medium was used with the supplementation of 40 g·L⁻¹ sodium formate (HCOONa·2H₂O).

(ii) Please provide evidences that the S-TCA truly contributed to the enhanced formate consumption. All the S-TCA genes need to be properly expressed, and the metabolites throughout the S-TCA should have certain level of concentrations.

Reply: According to the reviewer’s suggestions, we compared the transcriptional levels of all the genes in the S-TCA cycle of the wild-type and S-TCA-1.0 strains. As shown below (Figure 4), all these genes could be properly expressed in the S-TCA-1.0 strain, and moreover, most of them exhibited higher transcriptional levels in S-TCA-1.0 relative to the wild-type strain.

In addition, we measured eight crucial metabolites (glycine, serine, hydroxypyruvate, glycerate, phosphoenolpyruvate, α -Ketoglutarate, succinic acid, and fumarate) in the S-TCA cycle of the wild-type and S-TCA-1.0 strains. As shown below (Figure 5), all these eight compounds could be detected, in which five showed higher concentration in S-TCA-1.0 relative to the wild-type strain, while the other three had no significant differences in concentration between the two strains

These findings strongly indicate that the S-TCA cycle truly exerts its function in *V. natriegens*, leading to enhanced formate metabolic flux. The figures have been added into the revised manuscript (Supplementary Fig. 7 and Supplementary Fig. 8).

Figure 5 Comparison of the metabolite levels in the S-TCA cycle between the S-TCA-1.0 and wild-type strains. The LBv2 medium was used with the supplementation of 40 g·L⁻¹ sodium formate (HCOONa·2H₂O). HP: hydroxypyruvate. α-KG: α-ketoglutarate. C: the control strain (wild-type). 1.0: the S-TCA-1.0 strain. Data are presented as the mean ± SD (*n* = 3). Error bars show SDs. Statistical analysis was performed by a two-tailed Student's *t*-test. *, *P* < 0.05; **, *P* < 0.01; ***, *P* < 0.001; versus the control strain. ns: no significance.

Reviewer #2 (Remarks to the Author):

Major Comments

1. The authors used sodium formate in this study and compared the effects on growth and transcription of *V. natriegens* under varying concentrations. Sodium is essentially required for growth of this organism but higher concentrations have negative effects on growth (Hoffart et al. <https://doi.org/10.1128/AEM.01614-17>). Therefore, it is not clear whether the observed effects derive from sodium, formate or both. **(i)** Please perform control experiments for growth in complex LBv2 medium and minimal medium where you apply the respective sodium concentrations without formate. Since sodium probably also plays an important role in energy conservation and transport mechanisms in this organism these experiments are essentially required. **(ii)** Please also apply such control experiments for the transcriptomic analysis. Of course, you can also use formic acid throughout the work. As stated in the introduction formic acid would be the substrate of choice. This is an important issue since the utilization of sodium formate creates a huge salt load that has to be handled on an industrial scale.

Reply: (i) Thanks for the reviewer's comments and suggestions. To examine whether high concentration of sodium has negative effect on the growth of *V. natriegens*, it was grown in the nutrient-rich LBv2 medium and the M9 minimal medium supplemented with different concentrations of sodium chloride (NaCl, containing the same Na⁺ moles as those of sodium formate).

As shown below (Figure 6A and B), with the gradually enhanced concentrations of NaCl in the media (both the LBv2 and M9 media), its inhibition on the growth of *V. natriegens* increased. However, such a negative effect was much less than that caused from sodium formate (Figure 6C and D). **Therefore, we think that the observed severe inhibition of**

sodium formate on the growth of *V. natriegens* should be mainly attributed to the effect of formate.

Figure 6 The influence of sodium formate or sodium on the growth of *V. natriegens*. (A, B) The cell growth in the LBv2 medium (A) and the M9 minimal medium (B) supplemented with different concentrations of NaCl. (C, D) The cell growth in the LBv2 medium (A) and the M9 medium (B) supplemented with different concentrations of sodium formate.

(ii) We also performed the comparative transcriptomic analysis of *V. natriegens* in the presence and absence of high concentration of NaCl (385 mM) according to the reviewer's suggestion.

Following NaCl addition to the culture, there were 196 and 250 genes significantly up-regulated and down-regulated, respectively, which were much less than those (755 and 606, respectively) with the addition of sodium formate. Additionally, 48 up-regulated genes and 71 down-regulated genes were found in both cases (with sodium formate and sodium chloride) (as shown below, Figure 7A). Noticeably, only 13 genes in the THF cycle, the reductive glycine pathway, and the serine cycle showed significantly changed transcription with the stress of sodium chloride (as shown below, Figure 7B), while this number with the stress of sodium formate was 46 (as shown below, Figure 7C). All these results suggest that the observed influence of sodium formate on the growth and global transcription of *V. natriegens* should be mainly attributed to the role of formate.

Figure 7 The global transcriptional changes of *V. natriegens* under the stress of sodium formate (HCOONa) or sodium chloride (NaCl). (A) The number of significantly up-regulated or down-regulated genes with the stress of sodium formate or NaCl. The cells were growth in the LBv2 medium supplemented with HCOONa or NaCl. (B and C) The significantly up-regulated or down-regulated genes involved in the tetrahydrofolate (THF) cycle, serine cycle, pyruvate formate-lyase (PFL) pathway and formate dehydrogenase (FDH)-mediated formate dissimilation pathway with the supplementation of NaCl (B) and HCOONa (C), respectively, in *V. natriegens*. The bottom box indicates fold change as gene expression with HCOONa (or NaCl) over gene expression without HCOONa (or NaCl).

2. In Fig. 1 a and b growth experiments are performed in LBv2 complex medium. In Fig. 2 b growth in minimal medium is shown, however, with much lower formate concentrations compared to the experiments with LBv2. **(i)** Please perform a set of experiments in minimal medium with varying formate concentrations. Probably, the claimed (l. 71 f) high tolerance is less pronounced in minimal medium. If so, this should be carefully described. **(ii)** Moreover, it would be interesting to see if in minimal medium with glucose the addition of formate also increases the biomass yield.

Reply: (i) According to the reviewer's suggestion, we examined the tolerance of the *V. natriegens* wild-type strain in the M9 minimal medium with 4 g·L⁻¹ glucose and varying formate concentrations. As shown below, when the strain was cultivated in the M9 minimal

medium, the addition of 288 mM formate ($30 \text{ g}\cdot\text{L}^{-1} \text{HCOONa}\cdot 2\text{H}_2\text{O}$) could completely inhibited cell growth (Figure 8A), showing lower formate tolerance compared with the level in the LBv2 medium (Figure 8B). Nonetheless, the strain still could consume 95 mM formate within 24 h ($3.96 \text{ mM}\cdot\text{h}^{-1}$) using the M9 medium initially supplemented with 192 mM formate (equivalent to $20 \text{ g}\cdot\text{L}^{-1} \text{HCOONa}\cdot 2\text{H}_2\text{O}$) (Figure 8C and D).

This result has been added into the revised manuscript (Supplementary Fig. 1; Page 5, Line 72-78)

Figure 8 The formate tolerance and consumption of the *V. natriegens* wild-type strain in the M9 minimal medium. (A) The growth of *V. natriegens* in the M9 medium with varying formate concentrations. $4 \text{ g}\cdot\text{L}^{-1}$ glucose was added into the medium. (B) The growth of *V. natriegens* in the LBv2 medium with varying formate concentrations. (C) The residual formate in the M9 medium after 24 h cultivation of *V. natriegens* with different initial concentrations of formate (D) The formate consumption of *V. natriegens* after 24 h cultivation with different initial concentrations of formate.

(ii) The above result (Figure 8A) also answered the reviewer's question regarding whether the addition of formate in the minimal medium with glucose can increase the biomass yield. With the gradually increased formate concentrations, the growth rate and the final biomass of the strain decreased step by step. Therefore, it seems that no promotion effect of formate on the biomass yield of the *V. natriegens* wild-type strain grown in the minimal medium with glucose.

3. The authors applied ALE to improve the formate consumption of the engineered strain S-TCA-1.0. **(i)** Please re-sequence the evolved strain S-TCA-2.0 and provide the mutations in the genome. This information is also essential to understand the mechanism of formate assimilation. **(ii)** For the ALE approach, the question arises whether the strain is evolved to tolerate high sodium or formate concentrations or both.

Reply: (i) According to the referee's suggestion, we chose three isolates of S-TCA-2.0 for genomic sequencing, aiming to find crucial mutations in this evolved strain.

A total of nine mutated genes occurred in all the three isolates of S-TCA-2.0 relative to the S-TCA-1.0 strain (As shown below, Figure 9). Among them, *rpoS*, *ZapC*, and *actP*, encoding a general stress responsible regulator, a division protein, and an acetate permease, respectively, are commonly observed to be mutated in ALE experiments. Thus, the mutations of these three genes are unlikely to be specifically linked to the phenotypes (high formate tolerance and consumption) of S-TCA-2.0. Of note, we found a mutation (A44E) in *fumA*, encoding fumarate hydratase that converts fumarate toward malate, a main flux branch point of the S-TCA cycle. The mutation in this enzyme may reduce the diversion of metabolic flux from the S-TCA cycle. Another potentially key mutation is in *sdhC*, coding for a cytochrome subunit of succinate dehydrogenase (catalyzing the conversion of succinate to fumarate in the TCA cycle), whose inactivation may change the metabolic flux of TCA cycle. For the remaining four mutated genes, there is currently no clear clue linked to formate consumption. Future work will help elucidate whether the mutations in these genes indeed contribute to the improved performance of S-TCA-2.0 in formate consumption.

These data and the related elucidation have been added into the manuscript (Fig. 3d; Page 13, Line 192-204).

Figure 9 The nine mutated genes occurred in all the three isolates of the S-TCA-2.0 strain.

(ii) According to the reviewer's suggestion, we tested the tolerance of the S-TCA-2.0 strain to high concentration of sodium or sodium formate. As shown below (Figure 10), the strain could grow in the LBv2 medium with the addition of 769 mM (80 g·L⁻¹) sodium formate (HCOONa·2H₂O) or the same molar concentration of NaCl, demonstrating that S-TCA-2.0 can tolerate high concentration of sodium or sodium formate.

Figure 10 The tolerance of the *V. natriegens* S-TCA-2.0 strain to high sodium or sodium formate concentrations.

4. Since the strain with deleted *fdh*, *ftl*, and *pfl* genes still shows significant formate consumption and growth (Fig. 2 b), it is not clear how formate assimilation works in *V. natriegens*. Please provide flux analysis to get a more comprehensive picture of the formate metabolism. This is important because the phenotype of the *fdh* deletion strain is surprising (Fig. 2 b). One would assume that this reaction is essential for growth on formate.

Reply: Sorry for not describing this part clearly in the manuscript. The medium used in Fig. 2b was the M9 minimal medium (containing 4.0 g·L⁻¹ glucose) supplemented with 5.0 g·L⁻¹ sodium formate (HCOONa·2H₂O), because *V. natriegens* can not grow using formate as the sole carbon source. Thus, the deletion of *fdh* will not completely block the growth of *V. natriegens* in this medium, although this gene is essential for strain growth on formate. An explanation has been added into the figure legend (Page 44).

To better understand how formate assimilation works in *V. natriegens*, we performed ¹³C-tracer experiments. Based on the genomic information, the potential formate assimilation pathways in *V. natriegens* include the THF cycle and the Pfl-mediated pyruvate synthesis (as shown below, Figure 11A). Thus, we measuring the ¹³C labeling patterns of pyruvate and methionine (a proteinogenic amino acid derived from THF cycle) in the wild-type *V. natriegens* strain grown in the M9 medium (containing 4.0 g·L⁻¹ glucose and 5.0 g·L⁻¹ ¹³C-labelled sodium formate). As shown below (Figure 11B), the ¹³C-labelled methionine (M1) accounted for 64.7% of total methionine, indicating the incorporation of the methyl group propagated from ¹³C-formate. In contrast, the ¹³C-labelled pyruvate accounted for only 1.6% of total pyruvate. These results let us to conclude that the THF cycle play a major role in the formate assimilation of *V. natriegens* and the contribution of Pfl is very small. This finding is also consistent with the result in the manuscript that the deletion of the *pfl* gene had almost no influence on the formate consumption of *V. natriegens* (Fig. 2b in the manuscript).

Figure 11 The ^{13}C labeling patterns of pyruvate and methionine in the wild-type *V. natriegens* strain.

The cells were grown in the M9 medium (addition of $4 \text{ g}\cdot\text{L}^{-1}$ glucose $5 \text{ g}\cdot\text{L}^{-1}$ ^{13}C -labelled sodium formate).

5. Indigoidine production: **(i)** Please provide the maximal theoretical and achieved product yield on formate. Please consider for the calculation the requirements for reduction equivalents. **(ii)** After 60 h, indigoidine is degraded although formate is still present. What is the reason? **(iii)** Can you provide information about the production in minimal medium?

Reply: (i) Thanks for the reviewer's suggestion. The maximal theoretical indigoidine yield on formate was shown below, in which the requirements for reduction equivalents has been considered.

According to this equation, 8 mol formate (molecular weight 45) can generate 1 mol indigoidine (molecular weight 248), giving a **product yield of 0.69 g/g formate**.

But if formate is used as the sole carbon source, the 14 NAD(P)H and 2 CO_2 in the equation will have to come from formate dissimilation; thus, it will be 24 mol formate generating 1 mol indigoidine, giving a **product yield of 0.23 g/g formate**.

(ii) According to the previous report (Hui, et al. Indigoidine biosynthesis triggered by the heavy metal-responsive transcription regulator: a visual whole-cell biosensor. Applied Microbiology and Biotechnology, 2021, 105: 6087-6102.), **indigoidine in the aqueous phase is unstable and completely faded after a 16-h preservation at room temperature (RT) (as shown below)**. In contrast, the blue color degree was almost unchanged after a 16-h preservation at 4 °C. Thus, we think that the decreased concentration of indigoidine in the fermentation broth of the S-TCA-2.0-IE strain after 60 h should be attributed to indigoidine degradation.

(iii) We found that the growth of the S-TCA-2.0-IE strain was very poor in the M9 minimal medium (with the addition of 4 g·L⁻¹ glucose, 0.25 g·L⁻¹ glycine, and 5 g·L⁻¹ sodium formate). After 24 and 48 h of cultivation, only very small amount of indigoidine could be produced (as shown below). Therefore, it seems that the minimal medium is unsuitable for indigoidine production by the S-TCA-2.0-IE strain.

Reviewer #3 (Remarks to the Author):

Vibrio natriegens has received large attention for its favorable properties of fast growth and uses of syngas and C1 carbons. This manuscript first presents a through omics study of *V. natriegens* grown on formate. Metabolic engineering approach was applied to generate a formate sink by linking formate assimilation with a fused cycle of the serine and the TCA cycles, resulting in a strain S-TCA-1.0. Adaptive laboratory evolution of S-TCA-1.0 was done to create a high-performance strain which can grow in a medium with sodium formate as high as 140 g/L with the consumption of 78.9 g·L⁻¹ sodium formate within 24 h following initial supplementation of 85 g·L⁻¹ sodium formate in the medium. The cellular adaptation of the wild-type strain under formate stress was found to be much slower than that of S-TCA-1.0, suggesting that the reconstituted formate metabolic pathways in *V. natriegens* (S-TCA-1.0) facilitated the strain evolution to adapt to high formate stress. The strain S-TCA-2.0 was further engineered for efficient synthesis of indigoidine from glutamate first and then from formate. This work could be of significant interest and impact if the major issues as stated

below could be properly addressed.

Major issues:

1. Although the authors claimed that a high formate consumption rate was achieved in the engineered *V. natriegens*, the novelty and experimental design of this work need more clarification. Fundamentally, they did all the experiments on complex medium-LB containing yeast extract, peptone and others; this makes the mechanism of formate metabolism obscure and intractable. For example, as shown in Fig. 2c+4a, glycolysis contributed to the formation of the precursors for formate assimilation but it is not clear what level of glycolytic flux is required to stimulate or maintain a formate assimilation?

Reply: Thanks for the reviewer's suggestion. To clarify the potential influence of glycolysis on formate consumption, we added 5 or 10 g·L⁻¹ glucose into the LBv2 medium (containing 80 g·L⁻¹ HCOONa·2H₂O) and examined the influence on growth and formate consumption of the *V. natriegens* S-TCA-2.0 strain. As shown below (Figure 12), although the addition of glucose could enhance the cell growth, it had no influence on formate consumption. This finding indicates that the formate assimilation of the engineered S-TCA-2.0 strain is less dependent on the precursors from glycolysis.

Figure 12 The influence of glucose on the growth and formate consumption of the *V. natriegens* S-TCA-2.0 strain grown in the LBv2 medium.

2. Can the strain grow in a minimal medium with formate as the only carbon source or a co-substrate (even though the growth rate may be low)? What is the carbon yield and how much formate is assimilated into the products as well as the biomass? These information/data are very important to evaluate the real potential of *V. natriegens* as a C1 utilizer.

Reply: We found that the engineered strain S-TCA-2.0 cannot grow in the M9 minimal medium using formate as the only carbon source, but can grow in this medium using glucose and sodium formate as the co-substrates as well as adding a small amount of glycine.

Therefore, we cultivated the S-TCA-2.0 strain in the modified M9 medium and then collected samples for ¹³C isotopomer analysis to determine how much formate can be assimilated. In brief, the *V. natriegens* S-TCA-2.0 cells were grown in the modified M9 medium supplemented with 2.0 g·L⁻¹ glucose, 0.25 g·L⁻¹ glycine, and 10 g·L⁻¹ ¹³C-labeled sodium formate (HCOONa·2H₂O). After 48 h of cultivation, 1 mL of the grown cells were harvested and washed twice with the double distilled water. Next, the samples were freeze-dried and used for determining ¹³C contents by an isotope ratio mass spectrometer coupled with an elemental analyzer.

As shown below (Table 1), based on the specific consumption rates of formate, glucose, and glycine, the specific formate assimilation rate was calculated to be 43.3 mg·gDCW⁻¹·h⁻¹, accounting for 12.1% of the total consumed formate. Considering that a portion of formate would be assimilated into the metabolites that that can be secreted outside the cell, the real formate assimilation rate should be higher than 12.1%.

These data and the related elucidation have been added into the revised manuscript (Supplementary Table 2; Page 14, Line 207-213 in the text)

Table 1 Specific glucose, glycine, and formate consumption rates and ¹³C ratio in biomass of the *V. natriegens* S-TCA-2.0 strain..

Samples	Specific glucose consumption rate (mg·gDCW ⁻¹ ·h ⁻¹)	Specific glycine consumption rate (mg·gDCW ⁻¹ ·h ⁻¹)	Specific formate consumption rate (mg·gDCW ⁻¹ ·h ⁻¹)	¹³ C ratio in biomass*	Specific formate assimilation rate [§] (mg·gDCW ⁻¹ ·h ⁻¹)
No.1	292.4	53.6	379.8	0.078	42.7 (11.2%)
No.2	292.4	44.6	335.4	0.082	43.9 (13.0%)
Average	292.4	49.1 ± 4.5	357.6 ± 22.2	0.080 ± 0.002	43.3 ± 0.6 (12.1% ± 0.9%)

*The ¹³C ratio in biomass was obtained through the method described in Methods.

§ The specific formate assimilation rate was calculated by (carbon moles from formate)/(carbon moles from formate, glucose, and glycine) as previously described (Junho Bang, et al. *PNAS*, 2018, 115(40): E9271–E9279). The number in the brace was calculated by (specific formate assimilation rate)/(specific formate consumption rate), indicating how much formate was assimilated into biomass.

3. In Fig. 4c, the results of C13 labelling experiments are not clearly described. Only one sentence for this is completely not enough. "The result showed that all the detected amino acids were labeled by C13". How much formate goes into the amino acids? Are there any differences in terms of C13 composition, in different amino acids (some are derived from PEP, some from TCA intermediates)? It seems that a large fraction of amino acids is unlabeled.

Reply: Thanks for the reviewer's suggestion. We have added detailed elucidation of this figure in the manuscript, especially focusing on the difference in ¹³C labeled carbon ratio of different amino acids and the possible reasons (as shown below).

"To further validate the formate assimilation in S-TCA-2.0-IE, amino acids formed from the artificial S-TCA cycle (glycine, serine, isoleucine, lysine, methionine, aspartic acid, glutamate, and phenylalanine) and pyruvate (alanine, valine, and leucine) were analyzed through the isotopic labeling experiment using ¹³C-labelled formate. The result showed that all the detected amino acids were labeled by ¹³C (mostly, m + 1), in which serine and methionine contained higher proportions of ¹³C-labelled amino acids (45.7% and 39.4%, respectively), probably due to that these two amino acids are more closely linked to formate assimilation (THF cycle) compared with the others. Therefore, these results suggested that cells of S-TCA-2.0-IE were able to assimilate formate."

4. As the authors mentioned, the genome of *V. natriegens* has more than six formate dehydrogenase genes. So it has a strong formate dissimilation capacity to degrade formate into CO₂ instead of using formate as a carbon donor. The authors highlighted several times that *V. natriegens* could metabolize a high amount of formate within 24 hs (Fig. 2e, 3c), but how much formate is oxidized to CO₂ and how much is assimilated as the carbon source? If

most of the formate is converted into CO₂, what is the significant advantage to develop such a microbial chassis (CO₂-Formate-CO₂ futile cycle)?

Reply: According to the above data (Table 1), over 12.1% of the consumed formate can be assimilated into biomass components in *V. natriegens*. Although most of the consumed formate is supposed to be oxidized to CO₂ via the dissimilation pathway, it will contribute to energy metabolism in cells, thereby promoting the growth or the utilization of other co-substrates of *V. natriegens*. This is also why the addition of formate could significantly enhance the indigoidine production of *V. natriegens* grown in the nutrient-rich LBv2 medium (Fig. 4d in the manuscript).

Considering the efficient formate consumption rate (31.6 mM·h⁻¹, Fig. 3c in the manuscript) of the *V. natriegens* S-TCA-2.0 strain, the formate assimilation ratio of 12.1% (Table 1 shown above) would result in a formate assimilation rate of 3.8 mM·h⁻¹, which, to our knowledge, is still higher than the formate assimilation rates reported in other microbial hosts (Supplementary Table 1 in the manuscript). Additionally, further optimization of the formate metabolic networks should be able to enhance the formate assimilation ability of the *V. natriegens* S-TCA-2.0. Therefore, we think *V. natriegens* is a promising microbial chassis to be developed for formate utilization.

5. The authors should focus more on the identification of potential bottlenecks in the S-TCA cycle to stimulate formate assimilation. For example, what would be the outcome of overexpression of GCS or the serine cycle?

Reply: We agree with the reviewer's opinion. To explore this possibility, we overexpressed the genes in GCS and the serine cycle in the *V. natriegens* S-TCA-2.0 strain and then examined the influence on the formate consumption (Figure 13a). We found that the overexpression of the PN96_01075 and PN96_20940 gene led to increased and decreased formate consumption, respectively, while the other genes did not cause significant changes (Figure 13b). This finding indicates that the metabolic flux of the S-TCA cycle still has room for further improvement. The identification and elimination potential bottlenecks in the S-TCA cycle will help to optimize the formate utilization of S-TCA-2.0.

These results and the related elucidation have been added into the manuscript (Page 19, Line 291-296).

(B)

Figure 13 Influence of overexpressing the genes of the glycine cleavage system (GCS) and the serine cycle on the formate consumption of the S-TCA-2.0 strain. a, The overexpressed genes of GCS and the serine cycle. **b**, The influence of overexpressing these genes on formate consumption. The LBv2 medium was used with the addition of 80 g·L⁻¹ sodium formate (HCOONa·2H₂O). Data are presented as the mean ± SD ($n = 3$). Error bars show SDs. Statistical analysis was performed by a two-tailed Student's t -test. *, $P < 0.05$; ***, $P < 0.001$; versus the control strain.

6. In some aspects, the transcriptomics data are hardly understandable or even misleading, e.g. *pfl* is upregulated by more than 8.9-folds, but its deletion has no effect on cell growth and formate metabolism. Please explain it.

Reply: Sorry for not appropriately elucidating the transcriptomics data and phenotypic outcomes of *pfl*. We have made the following revisions.

(i) We replaced the sentence "These findings **strongly suggest** that the formate assimilation and dissimilation pathways mediated by these genes **contribute to** formate metabolism in *V. natriegens*." with "These findings **indicate** that the formate assimilation and dissimilation pathways mediated by these genes **may be associated with** formate metabolism in *V. natriegens*.", aiming to emphasize that the transcriptional changes of the *fdh* and *pfl* genes only indicate their potential relationship to formate metabolism (Line 110-112).

(ii) An explanation for the phenotypic outcomes of *pfl* deletion was offered: In contrast, the deletion of the *pfl* gene had no influence on cell growth and formate consumption, although this gene was significantly up-regulated with the supplementation of formate. This indicates the low catalytic activity of the PFL enzyme towards formate in *V. natriegens* (Page 9, Line 128-131).

7. In supplementary Table 3, *V. natriegens* $\Delta fdh \Delta ftl \Delta pfl$ was mentioned to have three *fdh* genes deletion, how about the remaining three?

Reply: Sorry for the mistake in writing. Actually, the *V. natriegens* $\Delta fdh \Delta ftl \Delta pfl$ strain had eight genes deleted simultaneously, including six *fdh* genes (PN96_05840, PN96_05845, PN96_05850, PN96_05880, PN96_21155, and PN96_22795), one *ftl* gene (PN96_20840), and one *pfl* gene (PN96_08455).

It has been corrected in the supplementary Table 3.

8. The role of formate for indigoidine formation is not clear. Formate was only shown to be incorporated into amino acids for cell growth. Carbon for indigoidine formation is from glutamate!

Reply: To examine whether formate can be incorporated into indigoidine, we analyzed the

ratio of ^{13}C -labelled carbon of glutamine (the precursor of indigoidine) in the S-TCA-2.0-IE strain which was grown in the LBv2 medium supplemented with $60\text{ g}\cdot\text{L}^{-1}$ of ^{13}C -labelled formate. As show below (Figure 14), the detected M0 and M1 signals accounted for 87.1% and 12.9% of total isotopes, respectively, indicating that the carbon atom in formate could be incorporated into indigoidine, although the proportion was not high. This finding let us to speculate that a large portion of the consumed formate in S-TCA-2.0-IE was oxidized to CO_2 (catalyzed by formate dehydrogenase) to generate reducing power or assimilated into biomass components, which may promote the cell growth or other carbon sources (i.e., tryptone and yeast extract in the LBv2 medium) to be more used for indigoidine synthesis.

This result and the related discussion have been added into the manuscript (Supplementary Fig. 11; Page 20, Line 307-314).

Figure 14 Relative abundance of ^{13}C -labelled glutamine in the S-TCA-2.0-IE strain. The cells were grown in the LBv2 medium supplemented with $60\text{ g}\cdot\text{L}^{-1}$ of ^{13}C -labelled sodium formate. Data are presented as the mean \pm SD ($n = 3$). Error bars show SDs.

Reviewers' Comments:

Reviewer #1:

Remarks to the Author:

The authors successfully addressed all my comments.

Reviewer #2:

None

Reviewer #3:

Remarks to the Author:

The revised version addressed all my comments and can be accepted for publication.

Point-by-point responses to the referees' comments

Reviewer #1 (Remarks to the Author):

The authors successfully addressed all my comments.

Reply: thanks for the reviewer's comment.

Reviewer #2:

[Editor: The reviewer provides his/her recommendation of publication suggestion in Remark to Editor section.]

Reply: thanks for the reviewer's recommendation.

Reviewer #3 (Remarks to the Author):

The revised version addressed all my comments and can be accepted for publication.

Reply: thanks for the reviewer's comment.